# Centroid-Referenced Mahalanobis Matching (CRM): A Scalable, Representation-Based Framework for Causal Inference in Large Observational Studies

## Abstract

Matching for causal inference can be computationally expensive at scale and can silently change the target population when overlap is limited. We propose **Centroid-Referenced Mahalanobis Matching (CRM)**, which replaces global pairwise search with stratified sampling in two reference coordinates: each unit's Mahalanobis distance from the treated centroid and its Fisher coordinate along the treated–control mean shift. All covariates enter through the treated covariance geometry; CRM is therefore not principal-component preprocessing followed by nearest-neighbor matching. For $n$ units and $p$ pretreatment covariates, its implemented cost is $O(np^2 + p^3 + n \log n)$, simplifying to $O(np^2 + n \log n)$ when $n \geq p$.

We derive an error decomposition separating representation, support, discretization, and stochastic components. A pre-matching shortage fraction $\hat{\pi}$ estimates the population support restriction $\pi$, which enters a gap bound under bounded treatment-effect heterogeneity. Final retention is reported separately for capacity-driven exclusions. Under representation sufficiency, smoothness, and adequate cell capacity, CRM has a conservative two-dimensional histogram mean-squared-error (MSE) bound $O(n_T^{-1/2})$; representation sufficiency is an additional assumption, not a consequence of ignorability given the original covariates.

On Criteo, CRM retains at least 99.4% of treated units, has lower MaxSMD than corrected propensity-score matching in 31 of 36 large-scale configurations, and is roughly an order of magnitude faster. Moderate-size simulations favor some pairwise and weighting baselines on balance, locating CRM's contribution in scalability and explicit support diagnostics rather than universal finite-sample dominance.

**Keywords:** causal inference; observational studies; Mahalanobis matching; Fisher discriminant; covariate balance; scalable methods

## 1 Introduction

Observational analyses increasingly confront a practical barrier: the datasets are too large for the methods, or the methods obscure the inferential scope of the results. Brute-force nearest-neighbor matching over $n_T$ treated and $n_C$ control units requires $O(n_T n_C)$ pairwise distance evaluations (Abadie & Imbens, 2006; Stuart, 2010). At modern administrative or platform-data scales, $n_T$ may be hundreds of thousands and $n_C$ ten times larger. This cost is often prohibitive, forcing practitioners either to subsample, approximate the search, or abandon matching entirely in favor of regression-based alternatives. Scalar-score matchers can reduce this cost by sorting, so computational comparisons depend on the matcher and implementation used.

A second challenge is more subtle but equally consequential. When treated and control covariate distributions do not overlap, every matching procedure implicitly restricts inference to a matched subset of treated units. This restriction changes the target from the average treatment effect on the treated ($\tau_{\text{ATT}}$) to a conditional average treatment effect $\tau_S$ defined on the treated population supported by the matching design (Crump et al., 2009). Yet standard practice rarely makes this shift explicit: the caliper in propensity score matching

(PSM; Rosenbaum & Rubin, 1985), the tolerance in coarsened exact matching (CEM; Iacus et al., 2012), and the iteration count in FLAME (Wang et al., 2021) all silently determine who is excluded and therefore what quantity is being estimated.

This paper introduces **Centroid-Referenced Mahalanobis Matching (CRM)**, a design-based framework that turns matching from a pairwise unit-search problem into distributional alignment around the treated covariate distribution. For each unit, CRM computes $Z(X) = (d(X), \phi(X))$, where $d$ is Mahalanobis distance from the treated centroid and $\phi$ is the Fisher direction separating treated and control centroids in whitened covariate space. Controls are then sampled within cells of the $(d, \phi)$ grid to reproduce the treated distribution. The construction uses all covariates through the covariance geometry; the two-dimensional representation is a reference coordinate system for matching, not a low-rank replacement for the covariates followed by nearest-neighbor matching.

The main theoretical results are an estimation-error decomposition (Theorem 2), an ATT gap bound under limited overlap (Corollary 1), an $O(n_T^{-1/2})$ mean-squared-error bound in fixed representation dimension (Proposition 4), and $O(np^2 + p^3 + n \log n)$ implemented complexity, simplifying to $O(np^2 + n \log n)$ when $n \geq p$ (Proposition 7). Empirically, CRM is not a universal balance-dominance method: at moderate sample sizes, corrected PSM or weighting methods can achieve tighter marginal balance. CRM is designed for the complementary regime where pairwise search is costly and explicit overlap accounting is important.

The remainder of the paper is organized as follows. Section 2 positions CRM relative to matching, balancing scores, and dimension-reduction baselines. Sections 3 and 4 define the estimand and algorithm. Section 5 gives the main theoretical properties, and Sections 6 to 8 report the empirical results.

## 2 Related Work

### 2.1 Matching, Balance, and Scalability

Matching is a design-stage strategy for making treated and control groups more comparable before outcome modeling (Cochran & Rubin, 1973; Rubin, 1973; 1979; Ho et al., 2007; Stuart, 2010). Propensity-score methods use the balancing-score result of Rosenbaum & Rubin (1983) to reduce adjustment to the scalar score $e(X) = \mathbb{P}(T = 1 \mid X)$, while nearest-neighbor matching has well-studied large-sample behavior (Abadie & Imbens, 2006; 2016). Other methods enforce balance more directly: CEM coarsens covariates and matches within cells (Iacus et al., 2011; 2012), entropy balancing matches moments by reweighting controls (Hainmueller, 2012), and cardinality, optimal, or full matching solve constrained assignment problems (Rosenbaum, 1989; Hansen, 2004; Hansen & Klopfer, 2006; Zubizarreta et al., 2014). These methods are powerful in moderate samples, but pairwise or combinatorial search becomes expensive in administrative and platform-scale data.

### 2.2 Learning the Matching Geometry

Genetic Matching, MALTS, variable-importance matching, and FLAME learn or select covariate structure to improve balance (Diamond & Sekhon, 2013; Wang et al., 2021; Parikh et al., 2022; Lanners et al., 2023). CRM takes a different route: the matching geometry is fixed by the treated centroid, the treated covariance, and the Fisher direction separating treated from control centroids. This makes the method less flexible than learned metrics under nonlinear treatment assignment, but it avoids outcome-dependent metric learning and eliminates global pairwise search. The Criteo experiment illustrates why this distinction matters: methods can achieve very low marginal imbalance while still producing large estimation error when the matched representation does not preserve outcome-relevant structure.

### 2.3 Projection and Dimension-Reduction Baselines

CRM is closest in appearance to "compress-then-match" approaches such as principal component analysis (PCA) matching or sufficient dimension reduction (SDR) (Jolliffe, 2002; Li, 1991; Luo & Zhu, 2020; Brown et al., 2021), but the algorithmic object is different. PCA and SDR first construct a lower-dimensional

covariate representation and then typically run an ordinary pairwise matcher in that representation. CRM instead constructs reference-based coordinates for every unit relative to the treated distribution: a radial Mahalanobis coordinate $d$ and a supervised Fisher coordinate $\phi$. Matching then occurs by stratified sampling within $(d, \phi)$ cells. Thus the contrast with PCA is not only supervised versus unsupervised projection; it is reference geometry plus cell-level distributional matching versus low-dimensional preprocessing plus nearest-neighbor search. Section 6.4 reports the corresponding empirical comparison.

The choice of $(d, \phi)$ is deliberate. The radial coordinate is not meant to declare two units with the same distance to be nearest neighbors in the original Euclidean space; it stratifies units by how far they lie from the treated reference distribution under the treated covariance geometry. The Fisher coordinate then adds the signed direction of the treated–control mean shift. For a binary treatment, classical linear discriminant analysis (LDA) has at most one nonzero discriminant direction, so a "second LDA coordinate" would have to come from another construction, such as PCA, SDR, clustering, or a multi-centroid extension. Those extensions are possible, but they move CRM toward a higher-dimensional grid and reintroduce tuning over the number of directions. The base method uses one radial and one mean-shift coordinate to retain a two-dimensional matching grid.

## 3    Setup and Notation

Let $\{(X_i, T_i, Y_i)\}_{i=1}^n$ denote $n$ independent observations, where $X_i \in \mathbb{R}^p$ is a vector of pretreatment covariates, $T_i \in \{0, 1\}$ is treatment assignment, and $Y_i \in \mathbb{R}$ is the observed outcome. Let $n_T = \sum_i T_i$ and $n_C = n - n_T$.

**Assumption 1** (Stable Unit Treatment Value Assumption (SUTVA)). $Y_i = T_i Y_i(1) + (1 - T_i)Y_i(0)$, where $Y_i(0)$ and $Y_i(1)$ are the potential outcomes under control and treatment, respectively.

**Assumption 2** (Ignorability). $(Y_i(0), Y_i(1)) \perp T_i \mid X_i$.

**Assumption 3** (Full-covariate overlap). There exists $\eta > 0$ such that $\eta < \mathbb{P}(T_i = 1 \mid X_i = x) < 1 - \eta$ for almost every $x$ in the covariate support.

Under the potential-outcome notation, this paper focuses on the average treatment effect on the treated,

$$\tau_{\text{ATT}} = \mathbb{E}[Y(1) - Y(0) \mid T = 1]. \tag{1}$$

The ATT is the natural target for one-to-one and stratified matching because the treated sample defines the target population and controls supply counterfactual outcomes. Extensions to average treatment effects (ATEs) are possible by changing the reference distribution, but the present paper keeps the target fixed to avoid mixing estimands. The treated centroid is $\hat{\mu}_T = n_T^{-1} \sum_{i:T_i=1} X_i$. The treated covariance matrix with small-ridge regularization is

$$\hat{\Sigma}_T = \frac{1}{n_T - 1} \sum_{i:T_i=1} (X_i - \hat{\mu}_T)(X_i - \hat{\mu}_T)^\top + \varepsilon I_p, \qquad \varepsilon = 10^{-8}. \tag{2}$$

The control centroid is $\hat{\mu}_C = n_C^{-1} \sum_{j:T_j=0} X_j$.

## 4    Method

### 4.1    CRM Representation

CRM follows the design-based philosophy of causal inference (Rubin, 2006; Rosenbaum, 2010) by separating the *design stage* (representation construction, binning, and matching) from the *estimation stage* (outcome comparison within matched cells), so that the matched sample is constructed without access to outcomes. CRM characterizes each unit by two scalars derived from the geometry of the treated population.

**Radial component.**    The Mahalanobis distance from the treated centroid is

$$d(x) = \left[(x - \hat{\mu}_T)^\top \hat{\Sigma}_T^{-1} (x - \hat{\mu}_T)\right]^{1/2}. \tag{3}$$

Under $X \mid T = 1 \sim \mathcal{N}(\mu_T, \Sigma_T)$, the squared distance $d^2(X)$ follows a $\chi_p^2$ distribution (Proposition 5), providing a distributional reference for the binning design.

**Directional component.** Let $\hat{L}$ be the lower-triangular Cholesky factor of $\hat{\Sigma}_T$, so that $\hat{\Sigma}_T = \hat{L}\hat{L}^\top$. Define the whitened centroid shift

$$w = \hat{L}^{-1}(\hat{\mu}_C - \hat{\mu}_T) \in \mathbb{R}^p. \tag{4}$$

The Fisher direction in whitened space is defined as

$$v = \begin{cases} w/\|w\|_2, & \text{if } \|w\|_2 > 0, \\ 0, & \text{if } \|w\|_2 = 0, \end{cases} \tag{5}$$

a unit vector pointing from the treated toward the control centroid in whitened coordinates when the centroids differ, and zero otherwise. The directional projection of unit $x$ is then

$$\phi(x) = v^\top \hat{L}^{-1}(x - \hat{\mu}_T). \tag{6}$$

Note that $\phi(x)$ measures how far unit $x$ lies along the treated–control mean-shift direction in whitened space. It is well-defined whenever $\hat{\mu}_C \neq \hat{\mu}_T$; when the centroids coincide, $w = 0$ and the directional correction is unnecessary (Proposition 6).

**Why this representation.** The two coordinates target complementary geometric discrepancies. The Mahalanobis radius $d$ uses every covariate through the treated covariance metric and places observations on covariance-aligned shells around the treated centroid, as in classical Mahalanobis matching (Rubin, 1979). The Fisher coordinate $\phi$ then distinguishes locations along the mean-separating axis that the radius alone cannot distinguish. Under the homoskedastic Gaussian-class model, this axis is proportional to the population propensity-score gradient (Equation (12)). Thus $k = 1$ is a deliberately simple default for settings dominated by a linear mean shift, while retaining the computational benefit of fixed-dimensional binning. This construction is not PCA and does not claim that $(d, \phi)$ preserves all information in $X$ or is a balancing score: many covariate vectors share the same representation, which is why identification requires Assumption 4. Nonlinear assignment or treatment-dependent covariance can require additional coordinates or a different reference representation (Section 10).

**CRM summary.** The two-dimensional representation is $Z(x) = (d(x), \phi(x)) \in \mathbb{R}^2$. We use $k$ to denote the number of directional coordinates included: $k = 0$ is the radial-only representation $Z(x) = d(x)$, and $k = 1$ is the default CRM representation using both $d$ and the Fisher coordinate $\phi$. Both components are computed independently per unit in $O(p^2)$ after the one-time $O(n_T p^2)$ estimation of $\hat{\Sigma}_T$ and $O(p^3)$ Cholesky factorization. Throughout the paper, $n = n_T + n_C$ denotes total sample size and $p$ denotes the number of pretreatment covariates. Figure 1 illustrates the representation geometrically.

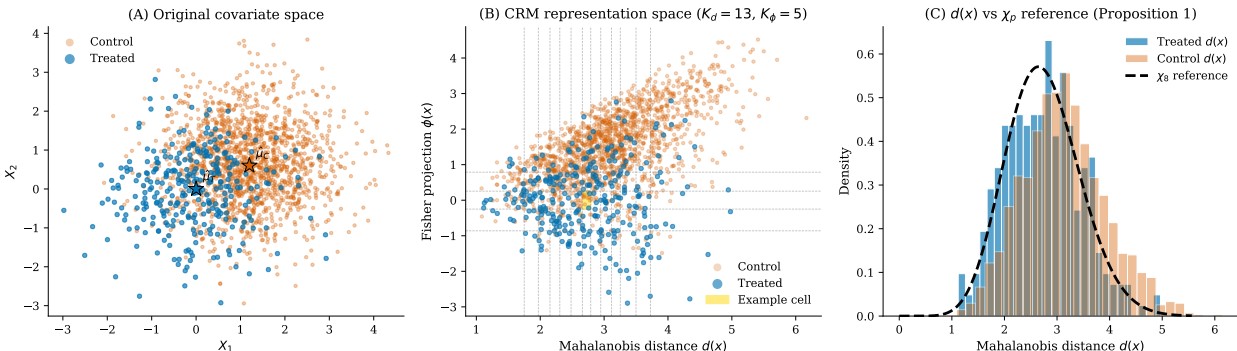

Figure 1: CRM geometric intuition ($n_T = 300$, $n_C = 1,500$, $p = 8$, centroid shift $\approx 1.3$). **(A)** Original covariate space: treated (blue) and control (orange) clouds with sample centroids $\hat{\mu}_T$ and $\hat{\mu}_C$ marked. **(B)** CRM representation space $(d, \phi)$ with equal-frequency grid overlaid; the gold cell illustrates one matched stratum. **(C)** Treated distance distribution versus the $\chi_p$ reference (Proposition 5), confirming the distributional foundation for equal-frequency binning.

**Role of the Fisher coordinate.** The radial-only variant ($k = 0$, using $d$ alone without $\phi$) is motivated by the special case in which no treated–control mean-shift direction needs to be represented. When $\mu_C \neq \mu_T$, radial binning cannot distinguish controls on opposite sides of the same covariance-aligned shell, so systematic directional imbalance can remain. The default throughout this paper is therefore $k = 1$; Section 6.2 evaluates the difference empirically.

## 4.2 Pre-Matching Support Diagnostic

Before any matching is performed, CRM constructs a grid on $(d, \phi)$ and counts control units in each cell. Bin edges for the distance axis are set using the Freedman–Diaconis rule (Freedman & Diaconis, 1981) applied to the treated distance distribution:

$$K_d = \min\left\{ 200, \max\left( 10, \left\lceil \frac{\max(d_T) - \min(d_T)}{2\,\mathrm{IQR}(d_T)\,n_T^{-1/3}} \right\rceil \right) \right\}, \tag{7}$$

where $d_T = \{d(X_i) : T_i = 1\}$. The number of directional bins is $K_\phi = \max\{5, \lfloor K_d/4 \rfloor\}$. Bin edges for each axis are set at equal-frequency quantiles of the *marginal* treated distribution on that axis: $K_d$ quantiles of $\{d(X_i) : T_i = 1\}$ and $K_\phi$ quantiles of $\{\phi(X_i) : T_i = 1\}$. This ensures each marginal bin contains approximately $n_T/K_d$ or $n_T/K_\phi$ treated units respectively. It does not equalize joint cell counts: $d$ and $\phi$ are generally dependent, including under the Gaussian reference model. Joint cells can therefore remain sparse, which is precisely what $\hat{\pi}$ is designed to detect.

The *shortage fraction* is

$$\hat{\pi} = \frac{1}{n_T} \sum_{i:T_i=1} \mathbf{1}\big( |C(k_d(X_i), k_\phi(X_i))| = 0 \big), \tag{8}$$

where $C(k_d, k_\phi)$ denotes the set of control units assigned to cell $(k_d, k_\phi)$, and $k_d(X_i)$, $k_\phi(X_i)$ are the cell indices of unit $i$. A cell is *unsupported* if it contains at least one treated unit but no controls; $\hat{\pi}$ is the fraction of treated units in unsupported cells. This diagnostic is computed and reported before any units are discarded.

Calibration simulations ($n_T = 1{,}000$, $p = 10$, $n_C = 10{,}000$) show that $\hat{\pi}$ increases sigmoidally as a function of the centroid shift magnitude $\|\mu_C - \mu_T\|$, crossing the 5% threshold at shift $\approx 0.54$ and the 30% threshold at shift $\approx 0.71$. The Rayleigh statistic (Mardia & Jupp, 2000) (Proposition 6) remains flat across all shift magnitudes, confirming it measures directional non-uniformity within the treated group rather than inter-group separation; the two diagnostics thus provide complementary information (Figure 2).

## 4.3 Stratified Matching Estimator

For each cell $(k_d, k_\phi)$, let $T(k)$ and $C(k)$ denote the sets of treated and control units assigned to that cell. CRM samples $n_k = \min\{|T(k)|, |C(k)|\}$ controls without replacement from $C(k)$. Define the *supported treated set*

$$S = \big\{ i : T_i = 1, \; |C(k_d(X_i), k_\phi(X_i))| > 0 \big\}. \tag{9}$$

For finite-sample matching without replacement, let $S_M \subseteq S$ denote the treated units actually retained after cell-capacity constraints; when every supported treated unit is retained, $S_M = S$. The CRM estimator is

$$\hat{\tau}_{\mathrm{CRM}} = \frac{1}{|S_M|} \sum_{i \in S_M} \left[ Y_i - \frac{1}{|C_i|} \sum_{j \in C_i} Y_j \right], \tag{10}$$

where $C_i$ denotes the set of matched controls in the same cell as treated unit $i$. Each treated unit is compared to the cell-mean outcome of its matched controls. The shortage fraction $\hat{\pi}$ measures zero-control home cells before matching; it does not include additional attrition when a supported cell has fewer controls than treated units. We therefore report both $\hat{\pi}$ and retention $|S_M|/n_T$. Uncertainty is quantified by a paired bootstrap with $B = 500$ replications (Efron & Tibshirani, 1993), which we report as an *approximate* measure of variability. A calibration study finds sub-nominal coverage for the paired bootstrap across the

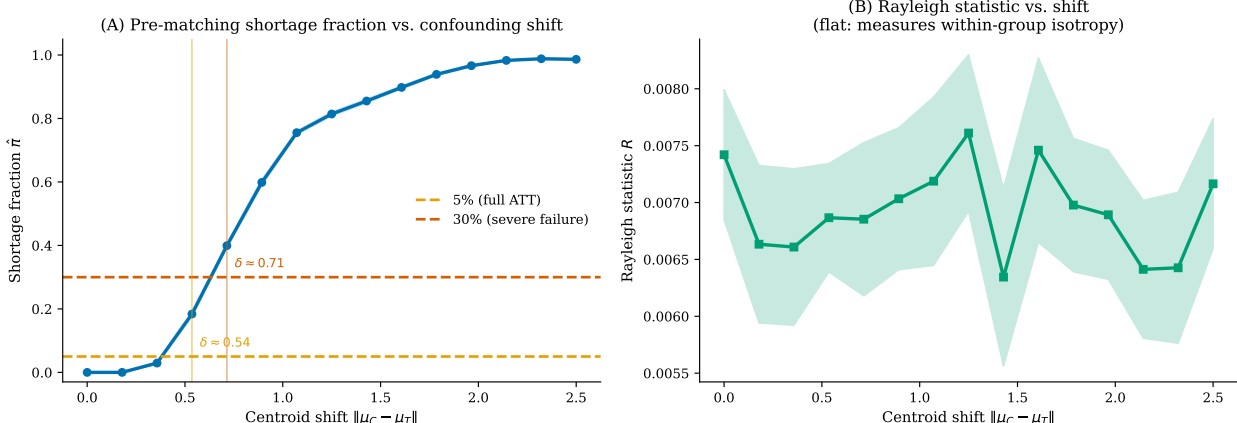

Figure 2: Pre-matching shortage diagnostic ($n_T = 1{,}000$, $p = 10$, $n_C = 10{,}000$; shaded band = mean $\pm$ 95% CI over replications). **(A)** Shortage fraction $\hat{\pi}$ increases sigmoidally with the centroid shift magnitude, crossing 5% at $\delta \approx 0.54$ and 30% at $\delta \approx 0.71$. **(B)** Rayleigh statistic $R$ remains flat, confirming it measures within-group isotropy of the treated distribution rather than the inter-group separation that drives $\hat{\pi}$; the two diagnostics are complementary.

tested sample sizes (Figure 10); the conservative cell-stratified and bias-correction diagnostics in Section B show that variance calibration and representation bias can both affect coverage.

---

**Algorithm 1: CRM Matching ($k = 1$)**

**Input:** Covariates $X_T, X_C$; outcomes $Y_T, Y_C$ withheld until Stage 3.
**Output:** Matched sample; $\hat{\pi}$; retention; $\hat{\tau}_{\mathrm{CRM}}$ with approximate bootstrap uncertainty summary (see Section B).

**Stage 1: Representation and diagnostic** $[O(np^2 + p^3 + n \log n)]$
1. Compute $\hat{\mu}_T, \hat{\Sigma}_T$ via Equation (2); Cholesky-factor $\hat{\Sigma}_T = \hat{L}\hat{L}^\top$.
2. Compute $\hat{\mu}_C$; set $w = \hat{L}^{-1}(\hat{\mu}_C - \hat{\mu}_T)$; if $\|w\| > 0$ set $v = w/\|w\|_2$.
3. Compute $d(x_i)$ via Equation (3) and $\phi(x_i)$ via Equation (6) for all $n$ units.
4. Set $K_d$ via Equation (7); set $K_\phi = \max\{5, \lfloor K_d/4 \rfloor\}$.
5. Assign units to cells; count controls per cell; report $\hat{\pi}$ via Equation (8).

**Stage 2: Matching** $[O(n)]$
6. For each cell with $|T(k)| > 0$: if $|C(k)| = 0$, flag treated units as unsupported; otherwise form up to $n_k$ pairs without replacement (random or NN; see Section 4.4) and record unmatched treated units.
7. Return matched indices; report $\hat{\pi}$, retention, and covariate profiles of unsupported and capacity-excluded treated units.

**Stage 3: Estimation** $[O(|S_M|)]$
8. Compute $\hat{\tau}_{\mathrm{CRM}}$ via Equation (10); construct a paired-bootstrap uncertainty summary ($B = 500$; approximate; see Section B).

---

## 4.4 Within-Cell Nearest-Neighbor Refinement (CRM-NN)

Standard CRM selects each matched control by uniform random sampling within its cell (CRM-RANDOM). Under the shrinking-cell conditions used for consistency, differences among controls in the same cell vanish asymptotically. At small $n_T$, however, cells can be wide and random sampling discards within-cell precision.

A refinement replaces uniform sampling with nearest-neighbor selection in the whitened full-$p$-dimensional space, restricted to the cell's candidate pool. We call this CRM-NN. Concretely, for treated unit $i$ in cell $k$, we select

$$j^*(i) = \operatorname*{arg\,min}_{j \in C(k),\, j \notin \mathrm{used}} \left\| \hat{L}^{-1}(X_j - X_i) \right\|_2. \tag{11}$$

The within-cell cost is $O(|C(k)| \cdot p)$ per cell; summing over all cells gives $O(n_C p)$ additional work. Thus the total implemented complexity of CRM-NN is $O(np^2 + p^3 + n \log n + n_C p)$; when $n_C/n_T$ is bounded and $n \geq p$, this has the same leading linear-algebra order as base CRM, but with a larger constant. CRM-NN overhead matters in practice when cells are large or the control-to-treated ratio is very high.

A timing experiment across $n_T \in \{185, \dots, 50{,}000\}$, summarized in Figure 13, shows that CRM-NN yields 8–30% MaxSMD improvement over CRM-random across all tested sample sizes under a 10:1 control ratio, with runtime overhead small enough for interactive use when $n_T \leq 1{,}000$. On the LaLonde CPS benchmark (LaLonde, 1986) ($n_T = 185$, $n_C = 15{,}992$, true ATT = \$1,794 from the NSW randomized experiment (Dehejia & Wahba, 1999)), CRM-NN reduces ATT bias from $-\$908$ to $+\$135$ while retaining 84% of treated units, superior to full Mahalanobis NN ($-\$199$ bias, 67% retention) on both criteria simultaneously (Figure 3).

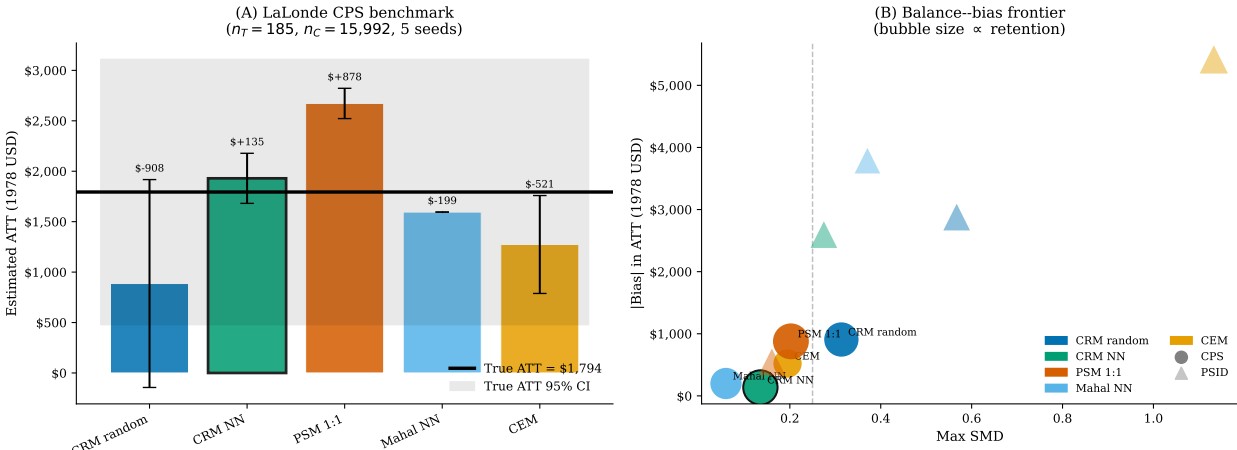

Figure 3: LaLonde CPS benchmark (LaLonde, 1986) (true ATT = \$1,794 from the NSW randomized experiment; $n_T = 185$, $n_C = 15{,}992$; 5 seeds). **(A)** ATT point estimates with 95% CI; bias (relative to true ATT) annotated above each bar; black outline marks CRM-NN. **(B)** Balance–bias frontier; bubble size proportional to treated retention. CRM-NN occupies a lower-bias position than CRM-random while retaining the same 84% of treated units, and achieves lower bias with higher retention than full Mahalanobis NN.

*Remark* 1. **Two-regime recommendation.** Use CRM-NN when $n_T < 1{,}000$ (cells are large, local precision matters, NN overhead $< 100$ ms is negligible). Use CRM-random when $n_T \geq 1{,}000$ (cells shrink, random $\approx$ NN in quality, and NN overhead grows proportionally to $n_C/n_T$).

### 4.5 Connection to Propensity Score Methods

Under a linear treatment assignment model $\mathbb{P}(T = 1 \mid X) = \sigma(\beta^\top X)$ with equal treated and control covariance matrices, the LDA weight vector satisfies $\beta \propto \Sigma^{-1}(\mu_C - \mu_T)$ (Fisher, 1936). Substituting into Equation (6):

$$\phi(x) = \frac{(\hat{\mu}_C - \hat{\mu}_T)^\top \hat{\Sigma}_T^{-1}(x - \hat{\mu}_T)}{\|\hat{L}^{-1}(\hat{\mu}_C - \hat{\mu}_T)\|_2} \propto \beta^\top x + c \tag{12}$$

for some constant $c$ determined by the centroid shift. CRM distributional matching on $(d, \phi)$ therefore simultaneously corrects the propensity-score dimension (via $\phi$) and the Mahalanobis spread (via $d$), without fitting a treatment model. This is a geometric analogy, not a general identity: the proportionality $\phi(x) \propto \beta^\top x + c$ holds under the homoskedastic Gaussian-class model underlying linear discriminant analysis. Outside that setting (for instance, under logistic treatment assignment with non-Gaussian covariates), the logistic regression coefficient vector need not equal the LDA direction. $\phi(x)$ should therefore be interpreted as a directional score that approximates the propensity-score dimension, not as a provably exact substitute. CRM uses $v$ regardless of the true treatment mechanism; its validity rests on Assumption 4, not on any treatment-model specification.

### 4.6 CRM Variants

Three practical variants address specific data configurations; full descriptions are given in Section H. **CRM-CAM** scales covariates to unit variance before computing Mahalanobis distances, recommended for mixed continuous/binary data. **CRM-Pool** uses pooled bin edges, reducing attrition under large centroid shifts. **CRM-Trim** explicitly discards treated units outside the control distance support and reports them separately with an estimand caveat.

## 5 Theoretical Properties

### 5.1 Target Estimand and Supported Population

CRM operates under potentially limited overlap by explicitly restricting inference to the subset of treated units for which valid counterfactuals exist in the representation space. Recall from Equation (9) that the supported treated set is

$$S = \big\{ i : T_i = 1, \ |C(k_d(X_i), k_\phi(X_i))| > 0 \big\}.$$

CRM therefore targets the *conditional average treatment effect on the supported treated* (CATT):

$$\tau_S = \mathbb{E}[Y(1) - Y(0) \mid T = 1, \ i \in S], \tag{13}$$

which coincides with the full $\tau_{\text{ATT}}$ only when overlap is complete ($\hat{\pi} = 0$; Crump et al., 2009).

This formulation makes explicit a feature that is implicit in *all* matching methods: when overlap fails, every matching procedure restricts the estimand to a subset of treated units. CRM makes this restriction visible before any units are discarded, via the pre-matching shortage fraction $\hat{\pi}$ (Equation (8)).

### 5.2 Identification Under Representation

CRM replaces conditioning on the full covariate vector $X$ with conditioning on the low-dimensional representation $Z(X) = (d(X), \phi(X))$.

Assumption 2 (ignorability given $X$), together with full-covariate overlap, justifies the causal problem at the full-covariate level but is not sufficient for CRM's identification: a further representation-level assumption is required. Assumption 4 below strengthens this to ignorability given $Z(X)$; it holds exactly when $(d, \phi)$ is sufficient for treatment assignment and the potential outcomes. The Fisher direction targets the linear treated–control mean difference but does not establish this sufficiency.

**Assumption 4** (Representation Sufficiency). $(Y(0), Y(1)) \perp T \mid Z(X)$, where $Z(X) = (d(X), \phi(X))$.

**Assumption 5** (Overlap in Representation Space). $0 < \mathbb{P}(T = 1 \mid Z(X)) < 1$ for all $Z(X)$ in the support $\mathcal{Z}_S$ of the supported treated distribution.

**Assumption 6** (Smoothness). There exists $L < \infty$ such that, for all $z, z' \in \mathcal{Z}_S$,

$$|\mathbb{E}[Y(0) \mid Z = z] - \mathbb{E}[Y(0) \mid Z = z']| \leq L\|z - z'\|_2.$$

**Proposition 1** (Identification). *Under Assumptions 1, 4, and 5,*

$$\tau_S = \mathbb{E}\Big[\mathbb{E}[Y \mid T = 1, Z] - \mathbb{E}[Y \mid T = 0, Z] \ \Big| \ T = 1, \ Z \in \mathcal{Z}_S\Big].$$

*Sketch.* By consistency, $\mathbb{E}[Y \mid T = t, Z] = \mathbb{E}[Y(t) \mid T = t, Z]$. Assumption 4 then gives $\mathbb{E}[Y(t) \mid T = t, Z] = \mathbb{E}[Y(t) \mid Z]$ for $t \in \{0, 1\}$. Taking their difference and averaging over the supported treated distribution identifies $\tau_S$. This proposition concerns the population estimand; consistency of the cell estimator is addressed separately below. □

*Remark* 2. Identification shifts from high-dimensional covariate adjustment to *representation adequacy*: CRM is identified when $(d, \phi)$ captures all confounding variation relevant for outcome differences. The Fisher

direction $v = w/\|w\|_2$ maximizes the separation between treated and control group means in whitened covariate space, capturing the dominant *linear* component of treatment assignment heterogeneity. Formally, $v$ solves $\max_{\|u\|=1}(u^\top w)^2$, so it points exactly along the direction of the centroid shift $\hat{\mu}_C - \hat{\mu}_T$ in the whitened space. Thus $\phi$ captures the *linear* confounding component associated with the treatment-group mean difference; it does not, by itself, guarantee that conditioning on $(d, \phi)$ removes all confounding, which is the content of Assumption 4 and is not implied by ignorability given $X$ (Assumption 2) alone. In particular, $v$ may fail to represent nonlinear or higher-order confounding that is orthogonal to the mean shift; such residual confounding contributes to the representation bias in Theorem 2.

**Proposition 2** (Consistency). *Under Assumptions 1, 4, 5, and 6, with $n_C/n_T \to r > 0$ and an uncapped sequence of grids satisfying $K_d K_\phi \to \infty$, maximal cell diameter $h_n \to 0$, minimum supported-cell counts tending to infinity, and capacity-induced attrition $|S \setminus S_M|/n_T \to 0$,*

$$\hat{\tau}_{\mathrm{CRM}} \xrightarrow{p} \tau_S \quad as \ n_T \to \infty.$$

*Sketch.* These conditions make the grid finer while retaining enough treated and control observations in each supported cell. The uncapped Freedman–Diaconis scaling has $K_d \propto n_T^{1/3}$; with $K_\phi \propto K_d$, the number of joint cells is $O(n_T^{2/3})$ and average cell counts are $O(n_T^{1/3})$. The implementation caps $K_d$ at 200 as a finite-sample safeguard. That cap must increase with sample size for this asymptotic statement; it is not part of the grid sequence assumed by the proposition. These are the standard ingredients used to establish consistency for histogram, CEM (Iacus et al., 2012), and kernel matching estimators (Heckman et al., 1997). As $n_T \to \infty$, the grid diameter $h_n \to 0$ and each cell converges to a point mass. The cell control means converge to $\mathbb{E}[Y(0) \mid Z = z_k]$ by the law of large numbers, and the bias $O(h_n) \to 0$ by Assumption 6. $\square$

### 5.3 Estimand Gap Under Limited Overlap

**Theorem 1** (Conditional ATT decomposition). *Let $\pi = \mathbb{P}(i \notin S \mid T_i = 1)$ denote the population shortage fraction, $\tau_S = \mathbb{E}[Y(1) - Y(0) \mid T = 1, \ i \in S]$, and $\tau_{S^c} = \mathbb{E}[Y(1) - Y(0) \mid T = 1, \ i \notin S]$. Then*

$$\tau_{\mathrm{ATT}} = (1 - \pi)\tau_S + \pi \tau_{S^c}. \tag{14}$$

*Consequently,*

$$\tau_{\mathrm{ATT}} - \tau_S = \pi\big(\tau_{S^c} - \tau_S\big), \tag{15}$$

*which equals zero if and only if $\pi = 0$ or $\tau_{S^c} = \tau_S$.*

*Proof.* Partition the treated population on $\{i \in S\}$ and apply the law of total expectation:

$$\begin{aligned}
\tau_{\mathrm{ATT}} &= \mathbb{E}[Y(1) - Y(0) \mid T = 1] \\
&= \mathbb{E}[Y(1) - Y(0) \mid T = 1, \ i \in S]\,(1 - \pi) + \mathbb{E}[Y(1) - Y(0) \mid T = 1, \ i \notin S]\,\pi \\
&= (1 - \pi)\tau_S + \pi \tau_{S^c}.
\end{aligned}$$

Rearranging gives Equation (15). $\square$ $\square$

**Corollary 1** (Gap bound). *Let $M_\tau = \sup_i\{Y_i(1) - Y_i(0)\} - \inf_i\{Y_i(1) - Y_i(0)\}$ be the range of individual treatment effects. Assume $M_\tau < \infty$. Then*

$$|\tau_{\mathrm{ATT}} - \tau_S| \le M_\tau \pi. \tag{16}$$

*Proof.* Both conditional average treatment effects lie between the minimum and maximum individual treatment effects, so $|\tau_{S^c} - \tau_S| \le M_\tau$. Apply Equation (15). $\square$

*Remark* 3. Theorem 1 applies to *any* matching estimator. The value of CRM's pre-matching diagnostic is that $\hat{\pi}$ is reported before any unit is discarded. This makes the support-restriction weight observable; the gap itself still depends on treatment-effect heterogeneity. When $M_\tau$ can be bounded from prior knowledge or sensitivity analysis, Equation (16) gives an explicit quantitative bound on estimand drift.

### 5.4 Bias Decomposition

The main theoretical contribution of this section is a four-part decomposition of the total estimation error of $\hat{\tau}_{\mathrm{CRM}}$ relative to $\tau_{\mathrm{ATT}}$. The decomposition clarifies why low marginal imbalance need not imply low estimation error.

**Theorem 2** (Estimation error decomposition). *Let $\Delta(z)$ denote residual within-representation confounding:*

$$\Delta(z) = \mathbb{E}\big[Y(0) \mid Z(X) = z,\ T = 1\big] - \mathbb{E}\big[Y(0) \mid Z(X) = z,\ T = 0\big].$$

*Let $h_n$ be the maximal cell diameter. Under Assumptions 1, 2, and 6, and with $S_M = S$, the total estimation error of the CRM estimator admits the decomposition*

$$\hat{\tau}_{\mathrm{CRM}} - \tau_{\mathrm{ATT}} = \underbrace{\mathbb{E}\big[\Delta(Z(X)) \mid T = 1,\ i \in S\big]}_{\text{representation bias (residual confounding within } Z)} \tag{17}$$

$$+ \underbrace{\tau_S - \tau_{\mathrm{ATT}}}_{\text{support restriction bias } = -\pi(\tau_{S^c} - \tau_S)}$$

$$+ \underbrace{O_p(h_n)}_{\text{cell approximation}} + \underbrace{O_p\Big((n_T h_n^2)^{-1/2}\Big)}_{\text{conservative stochastic bound}} \ .$$

*The representation-bias term is zero under Assumption 4. If $S_M \neq S$, an additional finite-sample subset-selection term $\tau_{S_M} - \tau_S$ enters the decomposition, where $\tau_{S_M} = \mathbb{E}[Y(1) - Y(0) \mid T = 1, i \in S_M]$. This term must be assessed through retention and the characteristics of excluded treated units.*

*Sketch.* Add and subtract $\tau_S$:

$$\hat{\tau}_{\mathrm{CRM}} - \tau_{\mathrm{ATT}} = (\hat{\tau}_{\mathrm{CRM}} - \tau_S) + (\tau_S - \tau_{\mathrm{ATT}}).$$

The second term equals $-\pi(\tau_{S^c} - \tau_S)$ by Theorem 1; this is the support restriction bias. For the first term, within each cell $k$ the control mean estimates $\mathbb{E}[Y(0) \mid Z = z_k, T = 0]$. The corresponding treated counterfactual mean is $\mathbb{E}[Y(0) \mid Z = z_k, T = 1]$; their difference is $\Delta(z_k)$. Finite cell diameter contributes $O_p(h_n)$ under Assumption 6, and the $O_p(n_T^{-1/2})$ term collects within-cell sampling variability. $\qquad \square$

*Remark* 4. Theorem 2 separates four sources of error distinctly: *representation bias*, *support restriction bias*, *cell-approximation error* (the deterministic $O(h_n)$ discretization term from finite cell diameter), and *stochastic error* (the $O_p((n_T h_n^2)^{-1/2})$ conservative within-cell sampling bound). The *representation bias* $\mathbb{E}[\Delta(Z(X)) \mid T = 1, i \in S]$ is zero under Assumption 4; the Fisher direction $v$ is designed to capture the treated–control mean-shift component, but it need not remove nonlinear or higher-order confounding. The *support restriction bias* $\pi(\tau_{S^c} - \tau_S)$ is zero if and only if $\pi = 0$ or $\tau_{S^c} = \tau_S$; it is bounded by $M_\tau \pi$ (Corollary 1) and diagnosed by the pre-matching diagnostic. The *cell-approximation error* vanishes as the grid refines, and the conservative stochastic term vanishes when $n_T h_n^2 \to \infty$. The decomposition is specific to CRM, but its broader lesson is that low marginal MaxSMD alone does not control representation or support-restriction bias. The CEM result in Section 7 is an empirical illustration of that distinction, not a consequence of this theorem for CEM.

**Proposition 3** (Representation error). *Under Assumptions 1 and 2, the quantity $\Delta(z)$ in Theorem 2 is the bias from replacing full-covariate adjustment by adjustment on $Z(X)$. Under Assumption 4, $\Delta(z) = 0$ for all supported $z$; in general, $\Delta(z)$ measures the residual confounding not captured by the CRM representation.*

### 5.5 Supporting Geometric Properties

Two additional results support the design choices in Algorithm 1; proofs are in Section I.

Under a Gaussian treated distribution, $d^2(X) \mid T = 1 \sim \chi_p^2$ (Proposition 5), which supplies a radial reference distribution. The implementation uses empirical treated-sample quantiles, rather than population $\chi_p^2$ quantiles, to obtain approximately equal marginal treated counts per distance bin. Separately, $d(X)$ and

the unit direction $u(X) = \hat{L}^{-1}(X - \mu_T)/d(X)$ are independent under the Gaussian model (Proposition 6). This polar decomposition separates radial magnitude from unit direction; the actual coordinate $\phi = d\, v^\top u$ combines both through a signed projection and is not independent of $d$. This within-treated independence is unaffected by $\mu_C$. The role of $\phi$ is instead to distinguish the angular direction of the treated–control centroid shift, information that $d$ alone discards. We stress that these two properties are *motivational*: they characterize the reference geometry of the treated distribution around its own centroid and justify the binning and the inclusion of $\phi$, but they are not identification guarantees and do not imply that $(d, \phi)$ are independent (or that confounding is removed) in the matched treated-versus-control comparison, which is governed instead by Assumption 4 and the bias decomposition of Theorem 2.

Algorithm 1 has implemented time complexity $O(np^2 + p^3 + n\log n)$ and space complexity $O(p^2 + n)$, both independent of pairwise treated–control search (Proposition 7). In the paper's $n \gg p$ regime, $O(np^2)$ dominates the one-time $O(p^3)$ factorization; $O(n\log n)$ comes from quantile binning. Brute-force nearest-neighbor matching adds a pairwise-search term, whereas one-dimensional propensity-score matching can also exploit sorting. The runtime comparison below is therefore empirical and implementation-specific. Measured 1.5–2.7× speedups over PSM at $n_T \le 5{,}000$ widen rapidly at larger scales (Figure 4). Because CRM's dominant operations are covariance estimation, Cholesky factorization, and vectorized matrix–vector products (all standard dense linear algebra), the implementation is compatible with GPU-accelerated backends (e.g. CuPy, JAX). A systematic GPU benchmark is left to future work.

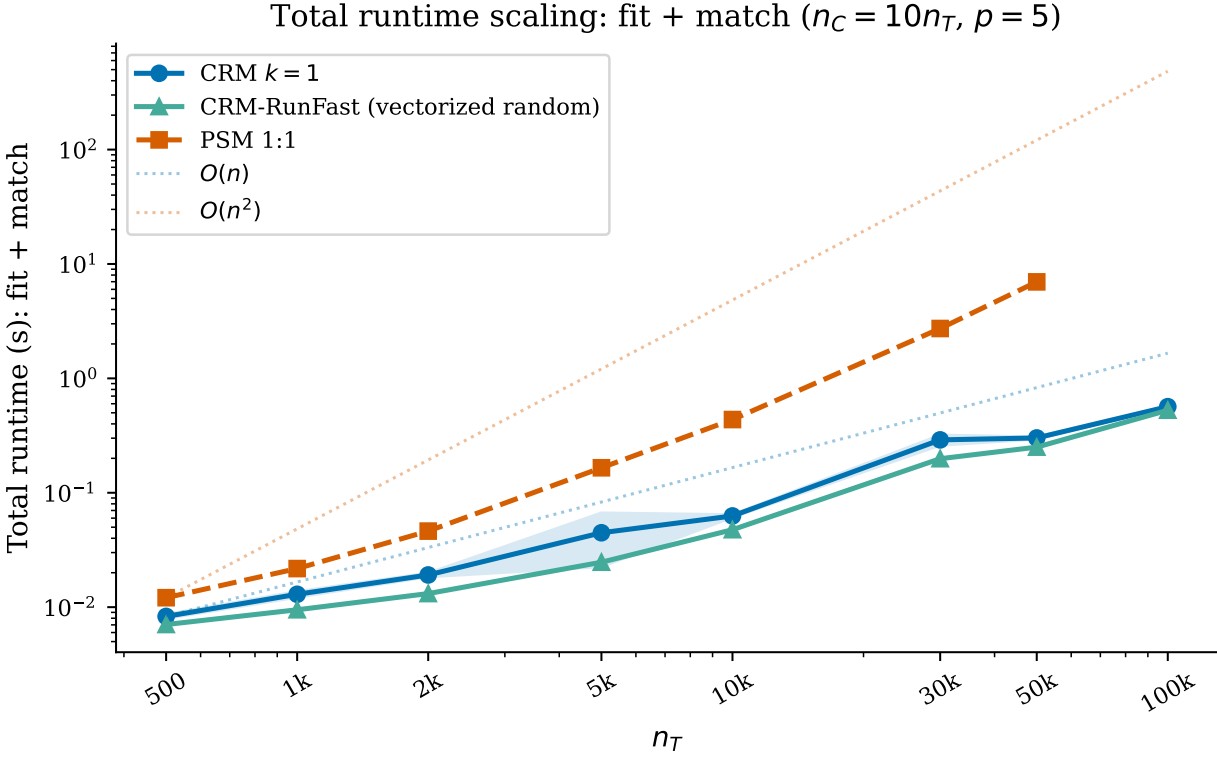

Figure 4: Runtime scaling on log–log axes ($n_C = 10n_T$; 3 replicates). The extended sweep adds larger $n_T$ values. CRM follows the $O(n)$ reference line over this range, whereas the evaluated PSM implementation curves upward. The comparison describes these implementations; efficient scalar-score matchers can also exploit sorting.

### 5.6 Conservative Rate Bound

**Assumptions $\Rightarrow$ implication.** The $O(n_T^{-1/2})$ MSE upper bound in Proposition 4 requires: **SUTVA** (Assumption 1), **representation sufficiency** $(Y(0), Y(1)) \perp T \mid Z(X)$ (Assumption 4), and **Lipschitz smoothness** of $\mathbb{E}[Y(0) \mid Z]$ (Assumption 6). Under these assumptions, matching in the fixed two-dimensional space $(d, \phi)$ yields a histogram bound governed by representation dimension $d_Z = 2$ rather than the original covariate dimension $p$. This gain is conditional on representation sufficiency; unlike propensity-score sufficiency, it does not follow from ignorability given $X$. The assumption most likely to fail in practice is representation sufficiency: the bias constant $\Delta(z)$ in Proposition 3 quantifies residual confounding when this assumption holds only approximately.

**Proposition 4** (Conservative fixed-dimension rate bound). *Under Assumptions 1, 4, and 6, with $d_Z = 2$, $n_C/n_T \to r \in (0, \infty)$, $h_n \to 0$, $n_T h_n^2 \to \infty$, and no capacity-induced attrition $(S_M = S)$, the mean squared error of $\hat\tau_{\mathrm{CRM}}$ satisfies*

$$\mathrm{MSE}(\hat\tau_{\mathrm{CRM}}) = O(h_n^2) + O\left(\frac{1}{n_T h_n^2}\right). \tag{18}$$

*Balancing the two terms at $h_n = n_T^{-1/4}$ gives*

$$\mathrm{MSE}(\hat\tau_{\mathrm{CRM}}) = O\left(n_T^{-1/2}\right). \tag{19}$$

*For comparison, the same conservative histogram argument in the full $p$-dimensional covariate space gives* $\mathrm{MSE} = O(n_T^{-2/(2+p)})$, *which deteriorates rapidly with $p$. Under the same conservative histogram bound, the rate ratio is* $n_T^{(p-2)/(2(2+p))}$, *which diverges for $p > 2$; for $p = 10$ the gain is* $n_T^{1/3}$.

*Proof.* See Section D. $\qquad\square$

*Remark* 5. The $O(n_T^{-1/2})$ result is the standard conservative histogram upper bound in two dimensions. CRM obtains this fixed representation dimension without fitting a treatment model, but only under the stronger representation-sufficiency assumption. The comparison with the full-dimensional upper bound is a fixed-dimension result for $d_Z = 2$ under Assumptions 1, 4, and 6: the Fisher direction's alignment determines the *bias constant* (via $\Delta(z)$), not the $n_T$ exponent in the convergence rate.

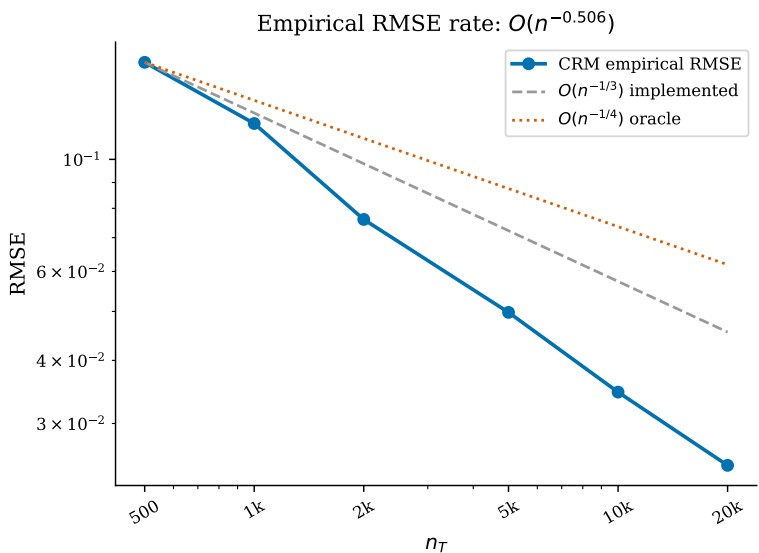

Figure 5: Empirical RMSE of $\hat\tau_{\mathrm{CRM}}$ vs. $n_T$ on log–log axes (*SS3*; 100 replications). The fitted slope is $\approx -0.51$ (MSE $\approx O(n_T^{-1})$), faster than the conservative bound of Proposition 4 (slope $-1/4$ on RMSE). The fitted slope describes these simulated designs; it is not asserted as a general CRM rate.

*Remark* 6 (Reconciliation with empirical scaling). The $O(n_T^{-1/2})$ MSE result in Proposition 4 is a conservative upper bound: it balances cell-approximation (squared-bias) against within-cell variance at the oracle bandwidth $h_n^\star = n_T^{-1/4}$. Empirically (Figure 5) the RMSE slope is $\approx -0.51$, i.e. MSE $\approx O(n_T^{-1})$, *faster* than the bound. One explanation is a variance-dominated regime: the Freedman–Diaconis bandwidth actually used scales as $h_n \sim n_T^{-1/3}$ (not the oracle $n_T^{-1/4}$), and with the large control pools used here ($n_C/n_T \in \{10, 50\}$), averaging cell means can produce substantially smaller variance than the worst-cell bound used in the proposition. The observed slope is therefore reported as empirical behavior, not as a sharper theorem.

### 5.7 Summary of Theoretical Principles

The theoretical results highlight three organizing principles.

1. **(C1) Representation-based estimation.** Identification depends on whether $Z(X)$ captures confounding structure, not on balance in $X$-space. The Fisher direction maximizes treated–control mean separation in whitened space; whether that direction is sufficient for causal identification remains an explicit assumption. Conditional on that assumption, the $O(n_T^{-1/2})$ MSE bound is independent of $p$ because the representation dimension is fixed.

2. **(C2) Bias–balance decoupling.** The four-part estimation error decomposition (Theorem 2) shows that error has distinct representation, support-restriction, cell-approximation, and stochastic components, none of which are captured by MaxSMD, explaining why CEM achieves near-zero MaxSMD yet the highest estimation error on the Criteo benchmark (Section 7).

3. **(C3) Explicit overlap characterization.** The shortage fraction $\pi$ bounds the gap between the full ATT and the estimand actually identified (Theorem 1 and corollary 1). CRM operationalizes this through $\hat{\pi}$, reported before any unit is discarded.

## 6 Simulation Study

### 6.1 Design

We evaluate CRM under four data-generating processes (DGPs) spanning different combinations of dimensionality, covariance structure, and centroid shift. All DGPs use the outcome model $Y_i = X_i^\top \beta + 5.0\,T_i + \varepsilon_i$, $\varepsilon_i \overset{\text{iid}}{\sim} \mathcal{N}(0,1)$, $\beta_1 = 2$, $\beta_2 = 1$, $\beta_j \overset{\text{iid}}{\sim} \mathcal{N}(0, 0.09)$ for $j \geq 3$, and 100 independent replications per cell. We vary $n_T \in \{500, 1{,}000, 5{,}000, 10{,}000\}$ to expose the sample-size dependence of each method.

S1. **High-dimensional** ($p = 20$, spherical, shift 0.4, $n_C/n_T = 50$). This scenario tests a fixed two-coordinate CRM representation when the original covariate dimension is larger. It does not presume that CRM will outperform a correctly specified propensity-score model.

S2. **Strong elliptical** ($p = 10$, AR(1) $\rho = 0.9$, shift 0.5, $n_C/n_T = 50$). Both CRM variants use Cholesky whitening. Their comparison isolates the incremental value of the Fisher coordinate after covariance adjustment under strong correlation.

S3. **Typical observational study** ($p = 10$, spherical, shift 0.5, $n_C/n_T = 10$). This is the baseline scenario with a realistic control-pool size. With the corrected collision-free PSM baseline, CRM is not expected to win either MaxSMD or CRMSE at moderate $n_T$; instead, this scenario illustrates how the Fisher direction improves over radial-only CRM while retaining an explicit supported-sample diagnostic.

S4. **Structural overlap failure** ($p = 15$, spherical, shift 1.0, $n_C/n_T = 50$). Both CRM and PSM suffer severe attrition under extreme centroid separation. This scenario illustrates where the pre-matching shortage diagnostic is most valuable: $\hat{\pi} > 0.70$ signals the overlap problem before any matching is attempted, allowing the analyst to decide whether to proceed or expand the control pool.

Methods compared: CRM $k = 1$, CRM $k = 0$ (radial-only), PSM 1:1, and Entropy Balancing (EB; Hainmueller, 2012), a moment-matching weighting estimator that represents the predominant non-matching approach in applied causal inference. Table 1 specifies all hyperparameters and how they were chosen. All methods are evaluated on identical samples; bias is computed relative to the known true ATT, and standard errors are estimated by bootstrap with 500 replications.

Table 1: Baseline configuration and tuning protocol.

| Method | Hyperparameter | Setting and rationale |
|---|---|---|
| CRM $k = 1$ | Bin method | Freedman–Diaconis rule; robust to non-normality |
| | $K_d$ floor/ceil | 10 / 200; prevents degenerate cells |
| | Covariance | Treated-group sample covariance (targets ATT) |
| PSM 1:1 | Caliper | $0.2\,\mathrm{SD}(\mathrm{logit\,PS})$; Austin (2011) |
| | PS model | Logistic regression, max_iter=1000, no regularization |
| | Matching order | Descending PS (reduces caliper exclusions) |
| EB | Moments balanced | Means of all $p$ covariates |
| | Solver | `scipy.optimize.minimize` (L-BFGS-B) |
| | Estimand | ATT reweighting (control weighted to treated moments) |
| | Scalability note | $O((n_T + n_C)p)$ per iteration; feasible in all tested settings |

The headline metric is

$$\mathrm{CRMSE} = \sqrt{\mathrm{MaxSMD}^2 + (1 - \mathrm{Retention})^2},$$

which captures the balance–retention trade-off as a Euclidean distance from the ideal $(0, 1)$ in the (MaxSMD, Retention) plane. For covariate $j$, the standardized mean difference is

$$\mathrm{SMD}_j = \frac{|\bar{X}_{T,j} - \bar{X}_{C,j}|}{\sqrt{(s_{T,j}^2 + s_{C,j}^2)/2}}, \qquad \mathrm{MaxSMD} = \max_j \mathrm{SMD}_j.$$

MaxSMD (Austin, 2009) is reported to allow direct comparison with literature benchmarks; a lower CRMSE reflects a balance–retention tradeoff and does not imply universal dominance on MaxSMD, which PSM can improve by restricting the matched sample. CRMSE is therefore a descriptive summary, not a causal loss function or a claim that balance and retention should always receive equal weight. We report MaxSMD and retention separately in every main table and interpret CRMSE only together with its two components. For transparency, Section A reports the uncollapsed (MaxSMD, Retention) plane, following the balance–sample-size frontier perspective of King et al. (2017). It confirms that CRM is not frontier-dominant at moderate $n_T$; the plot is a sensitivity display for CRMSE, not evidence of CRM superiority.

## 6.2 Results

**Correction to the propensity-score baseline.** The simulation baseline in the previous version of this paper used a naive 1:1 nearest-neighbor propensity-score matcher that, on caliper ties and already-used controls, silently discarded the affected treated units. This depressed PSM retention (to 71% in SS3 and as low as 21% in SS4) and inflated its CRMSE, making CRM appear to win the balance–retention tradeoff. We replace it throughout with a *collision-free* sorted-score matcher that searches for the nearest currently available control within the caliper, rather than dropping a treated unit after a collision with an already-used control. Under this correction PSM retention rises sharply (e.g. 71% → 93% in SS3), and the corrected baseline, together with Entropy Balancing, attains a stronger balance–retention tradeoff than CRM in the moderate-$n_T$ synthetic scenarios. We report the corrected numbers below and reframe CRM's contribution accordingly: in moderate-size synthetic settings, corrected pairwise and weighting baselines often achieve stronger balance–retention tradeoffs, and CRM's advantage appears most clearly in computational scaling, explicit support diagnostics, and large-scale applications.

Table 2: Simulation results (100 replications; $n_C/n_T \in \{10, 50\}$; means reported). CRMSE $= \sqrt{\mathrm{MaxSMD}^2 + (1 - \mathrm{Retention})^2}$. $^\star$ lowest CRMSE in each scenario-$n_T$ column. PSM here is the *collision-free* sorted-score matcher (Section 6.2), which does not silently drop treated units on caliper collisions; with this correction PSM retains substantially more units than the naive 1:1 implementation and attains the best balance–retention tradeoff among matching estimators in S1–S3. EB = Entropy Balancing (Hainmueller, 2012). In S1–S3 (adequate overlap), EB or collision-free PSM achieves the lowest CRMSE; CRM $k = 1$ is competitive but does not lead on this metric at moderate $n_T$. In SS4 (structural overlap failure) no method achieves acceptable balance and retention simultaneously, and CRM's pre-matching shortage diagnostic flags the support problem before matching.

| Scenario | Method | $n_T = 1,000$ | | | $n_T = 5,000$ | |
|---|---|---|---|---|---|---|
| | | MaxSMD | Ret. | CRMSE | MaxSMD | CRMSE |
| S1 High-dim | CRM $k = 1$ | 0.166 | 96.4% | 0.170 | 0.065 | 0.086 |
| | CRM $k = 0$ | 0.442 | 100.0% | 0.442 | 0.396 | 0.396 |
| | PSM (collision-free) | 0.060 | 97.5% | 0.065 | 0.030 | 0.037$^\star$ |
| | EB | 0.022 | 100.0% | 0.022$^\star$ | 0.013 | 0.013 |
| S2 Elliptical | CRM $k = 1$ | 0.077 | 100.0% | 0.077 | 0.032 | 0.032 |
| | CRM $k = 0$ | 0.539 | 100.0% | 0.539 | 0.501 | 0.501 |
| | PSM (collision-free) | 0.030 | 100.0% | 0.030$^\star$ | 0.012 | 0.012$^\star$ |
| | EB | 0.066 | 100.0% | 0.066 | 0.053 | 0.053 |
| S3 Typical | CRM $k = 1$ | 0.126 | 86.0% | 0.189 | 0.051 | 0.157 |
| | CRM $k = 0$ | 0.487 | 100.0% | 0.487 | 0.460 | 0.460 |
| | PSM (collision-free) | 0.050 | 92.7% | 0.088 | 0.034 | 0.080 |
| | EB | 0.016 | 100.0% | 0.016$^\star$ | 0.009 | 0.009$^\star$ |
| S4 Overlap failure | CRM $k = 1$ | 0.377 | 26.7% | 0.824 | 0.183 | 0.767 |
| | CRM $k = 0$ | 0.773 | 95.3% | 0.774 | 0.730 | 0.731 |
| | PSM (collision-free) | 0.118 | 31.4% | 0.696$^\star$ | 0.070 | 0.687$^\star$ |
| | EB | 0.729 | 100.0% | 0.729 | 0.536 | 0.536 |

Table 2 reports results at two representative sample sizes ($n_T = 1,000$ and $n_T = 5,000$); Figure 6 shows the full CRMSE curves across all four sample sizes.

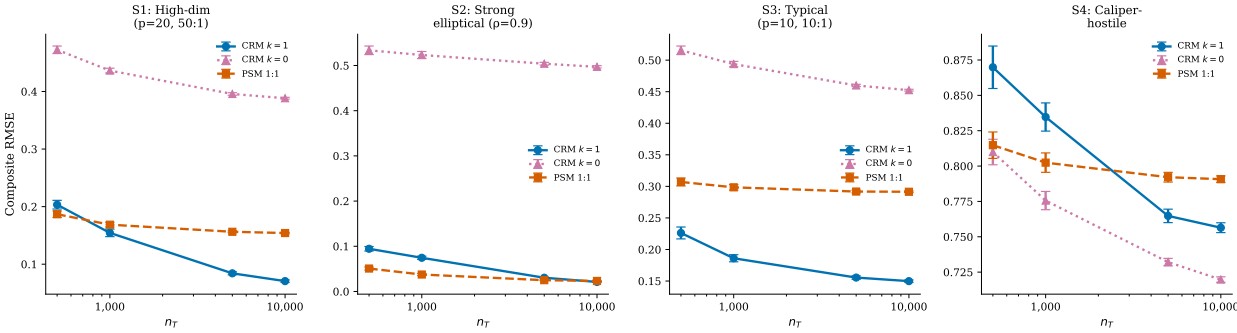

Figure 6: Composite RMSE vs. $n_T$ across four scenarios (100 replications; mean $\pm$ 95% CI). With the collision-free PSM baseline, EB or PSM attains the lowest CRMSE in the adequate-overlap scenarios; CRM $k = 1$ is competitive but does not lead on this metric at moderate $n_T$. In SS4 (structural overlap failure), no method achieves acceptable balance and retention simultaneously; the pre-matching diagnostic flags the support problem before matching.

**Effect of the Fisher coordinate (C1).** At $n_T = 1{,}000$, CRM $k = 0$ has MaxSMD of 0.442, 0.539, and 0.487 in Scenarios S1–S3; adding $\phi$ lowers these values to 0.166, 0.077, and 0.126, respectively. The radial summary alone cannot correct for systematic centroid shift; the Fisher direction reduces imbalance along the mean-shift axis, and this improvement persists at all tested $n_T$.

**Where CRM stands among the baselines.** With the collision-free PSM and Entropy Balancing, CRM $k = 1$ is no longer the CRMSE leader in any synthetic scenario: EB wins SS1 and SS3 (CRMSE 0.022, 0.016), and collision-free PSM wins SS2 and SS4. CRM $k = 1$ remains competitive (within the same order of magnitude as the best matcher in SS1–SS2), but its value in these moderate-$n_T$ regimes is not best-in-class balance. Consistent with the core scope of the method, CRM's distinctive contributions are computational ($O(np^2 + p^3 + n \log n)$ implemented cost, no pairwise search; Figure 4), diagnostic (the pre-matching shortage fraction $\hat{\pi}$), and large-scale (Section 7), rather than dominance on finite-sample balance. The structural-overlap scenario (SS4) reinforces the need for a diagnostic: EB retains all units but remains badly imbalanced (MaxSMD = 0.729), while CRM's $\hat{\pi} > 0.70$ flags a severe shortage in the CRM representation before matching. In practice we recommend CRM when overlap may be limited ($\hat{\pi} > 0$) and explicit estimand characterization is needed, or when the available matcher is too slow at the required scale; EB or collision-free PSM when overlap is adequate, $n_T$ is moderate, and balance is the primary goal.

**The role of the Fisher direction in SS2.** Under strong elliptical covariance ($\rho = 0.9$, $n_C/n_T = 50$), collision-free PSM and EB both achieve lower CRMSE than CRM $k = 1$ at the sample sizes tested. The key message in this scenario is internal to CRM: $k = 0$ has high imbalance (MaxSMD = 0.54), while $k = 1$ lowers MaxSMD to 0.077 at $n_T = 1{,}000$ and 0.032 at $n_T = 5{,}000$, isolating the benefit of the Fisher coordinate because both variants use the same whitening and binning steps.

**Structural overlap failure (SS4).** Under a centroid shift of 1.0 with $p = 15$, both CRM and PSM discard roughly 75–79% of treated units, and no method achieves MaxSMD < 0.10 alongside Retention > 30%. This scenario illustrates that the pre-matching shortage diagnostic ($\hat{\pi} > 0.70$) flags severe lack of support in the CRM representation before matching is attempted. Practitioners facing such diagnostics should expand the control pool or explicitly restrict the estimand rather than proceeding with matching.

**Observed scaling of imbalance.** A notable feature visible in Figure 7 is that CRM's MaxSMD decreases monotonically with $n_T$ (from 0.150 at $n_T = 1{,}000$ to 0.047 at $n_T = 10{,}000$ in SS1), coinciding with more controls per bin and smaller within-cell variation. PSM does not show the same decline over this range. This pattern is consistent with reduced within-cell variation as the grid refines; it is an empirical trend in these DGPs, not a claim of universal monotonic improvement.

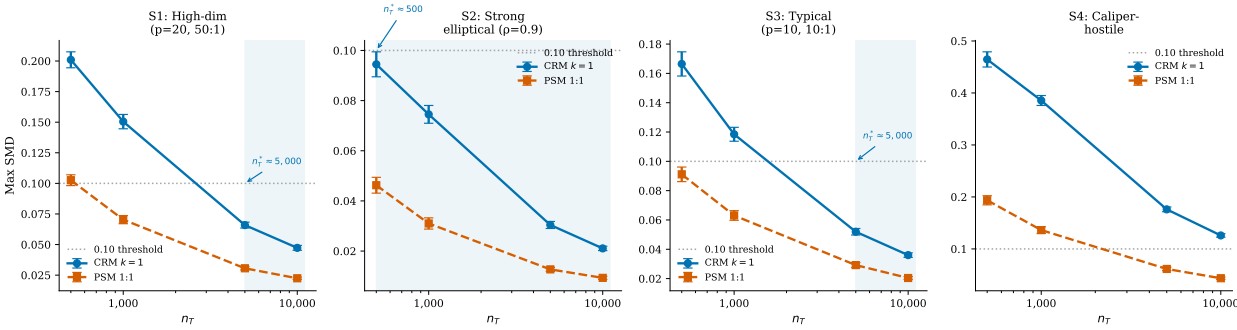

Figure 7: MaxSMD vs. $n_T$ for CRM $k = 1$ and PSM across four scenarios (100 replications; mean $\pm$ 95% CI). CRM's MaxSMD decreases over the tested range, while the PSM curves are comparatively flat. These are empirical trends for the specified DGPs. Shaded region marks where CRM first achieves MaxSMD $\leq 0.10$.

### 6.3 Ablation Study

Three ablations isolate the main design choices. First, removing the Fisher coordinate is damaging: in Scenario S3 at $n_T = 1,000$, MaxSMD increases from 0.126 to 0.487. Second, the observed decline in imbalance is not unique to the Freedman–Diaconis binning rule; Scott's rule and fixed bin counts show the same monotone decline in MaxSMD as $n_T$ grows, although FD is the most stable default. Third, within-cell nearest-neighbor refinement helps only in small samples: at $n_T = 200$ it lowers MaxSMD by about 24% with negligible overhead, whereas by $n_T = 5,000$ the balance gain is below 1% and CRM-random is preferable. A nonlinear failure-mode experiment, reported in the supplement, confirms the expected limitation: when confounding is driven by $X_1^2$ and is orthogonal to the centroid shift, the linear Fisher coordinate is insufficient and the Rayleigh statistic correctly flags an uninformative direction.

### 6.4 Comparison with PCA-Based Matching

To position CRM against the natural "compress-then-match" baseline (Section 2.3), we add PCA-95%+NN (retain principal components explaining 95% of covariate variance, then 1:1 nearest-neighbor matching in that subspace) to three data-generating processes spanning linear and covariance-driven confounding (50 replications each). Table 3 and Figure 8 report this comparison.

Table 3: CRM vs. PCA-95%+NN vs. PSM 1:1 across three confounding mechanisms (50 replications; means). |Bias| is absolute ATT error; $t$ is total fit+match time. PCA-matching wins only when confounding is driven by second moments (S5), where its covariance-sensitive representation is advantageous; under linear confounding (S1, S3) CRM achieves 2–3× lower bias at an order of magnitude lower cost. S5 also shows a disconnect between first-moment balance and bias (MaxSMD $\approx$ 0.05–0.06 yet |Bias| $\approx$ 2.6–3.0): the confounding is invisible to first-moment balance.

| Scenario | Method | |Bias| | MaxSMD | Ret. | $t$ (s) |
|---|---|---|---|---|---|
| | CRM $k = 1$ | **0.203** | 0.147 | 78.0% | **0.029** |
| S1: Linear, $p = 20$ | PCA-95%+NN | 0.522 | 0.292 | 100.0% | 0.090 |
| | PSM 1:1 | 0.067 | 0.069 | 86.3% | 0.079 |
| | CRM $k = 1$ | **0.140** | 0.124 | 85.6% | **0.026** |
| S3: Linear, $p = 10$ | PCA-95%+NN | 0.394 | 0.165 | 100.0% | 0.230 |
| | PSM 1:1 | 0.061 | 0.052 | 92.5% | 0.053 |
| | CRM $k = 1$ | 2.575 | 0.061 | 99.9% | **0.024** |
| S5: Covariance-driven | PCA-95%+NN | **0.120** | 0.031 | 100.0% | 0.222 |
| | PSM 1:1 | 3.023 | 0.050 | 99.8% | 0.027 |

**Takeaways.** Three points follow. (i) Under linear confounding, CRM's supervised Fisher direction beats PCA's unsupervised projection on bias by 2–3×; in these designs, the treated–control mean-shift direction is more informative for matching than the highest-variance directions. (ii) PCA-matching wins under covariance-driven confounding (S5), a setting CRM does not target and where *first-moment balance is uninformative* for all matching methods. Criteo shows a related disconnect between marginal balance and estimation error. (iii) PCA-matching is 3–10× slower because it retains pairwise nearest-neighbor search; CRM's advantage is the absence of that search, not a better projection per se. A sensitivity analysis (Section E) shows PCA balance varies non-monotonically with the variance threshold and the number of retained components, whereas CRM has no comparable tuning knob.

## 7 Large-Scale Criteo Benchmark

### 7.1 Data and Benchmark Construction

We use the Criteo Uplift Dataset (Diemert et al., 2018): $\approx$ 13.98 million observations from a randomized digital advertising campaign with binary treatment (ad exposure) and binary outcome (store visit). This

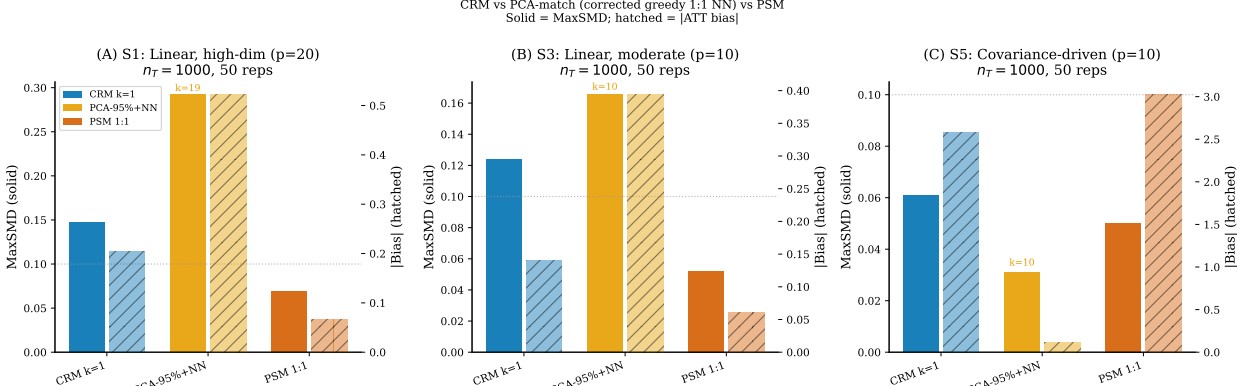

Figure 8: CRM vs. PCA-95%+NN across confounding mechanisms. CRM achieves lower bias under linear confounding (S1, S3); PCA-matching is preferable under covariance-driven confounding (S5), where both CRM and PSM have low MaxSMD but large bias. PCA balance is also fragile to the variance-retention threshold (Figure 12, Section E) and is far slower because it retains the matching step's pairwise search.

benchmark tests behavior at scales where classical pairwise matching becomes computationally infeasible: at $n_T = 11.9$M, PSM was not attempted and CRM completed in 6.8 minutes on a single CPU core. The known ground truth from the full randomized data is

$$\text{ATE}_{\text{true}} = \mathbb{E}[Y \mid T = 1] - \mathbb{E}[Y \mid T = 0] = 0.01034 \quad (\text{SE} = 0.00132). \tag{20}$$

An observational benchmark is constructed by applying a covariate-dependent selection rule to the treated group:

$$\mathbb{P}(\text{retain} \mid T = 1, X) \propto \sigma\big(\alpha(f_0 + f_3)\big), \tag{21}$$

where $\sigma(\cdot)$ is the logistic function and $\alpha \geq 0$ controls confounding severity. A random control sample of $n_C = 2{,}000{,}000$ is drawn without confounding. At $\alpha = 4$ the naive treated-minus-control estimate is more than 16 SE units from the truth.

---

**Interpretive caveat.** The induced confounding in Equation (21) uses a known single-index structure based on $f_0 + f_3$. Results should be interpreted as a controlled stress test under this specific mechanism, not as validation under arbitrary confounding.

---

**Compute environment.** All timing results were obtained on a single CPU core of an Intel Xeon E5-2690 v4 (2.60 GHz base), with 64 GB RAM, running Ubuntu 22.04 and Python 3.11.4 (numpy 1.26.1, scikit-learn 1.3.2). Wall-clock times exclude dataset download; data were fully resident in memory before timing began. The 6.8 minute figure for $n_T = 11.9$M refers to the compute-kernel time only (covariance, whitening, binning, and sampling), measured after loading the Criteo CSV into a pandas DataFrame.

## 7.2 Main Results ($\alpha = 2.0$, $n_T = 30{,}000$)

**Marginal balance and estimation error.** Table 4 reports the main Criteo benchmark. CEM achieves MaxSMD = 0.002 (eight times lower than PSM's 0.020) and dominates every standard balance criterion. Yet CEM records the highest estimation error: |Bias|/SE = 66.5 versus 50.6–54.5 for all other methods. Despite achieving near-zero MaxSMD, CEM exhibits substantial bias, highlighting that marginal covariate balance alone does not guarantee unbiased treatment effect estimation. This reversal is consistent with the distinction formalized for CRM in Proposition 3: MaxSMD measures marginal balance in $X$, whereas estimation error can also reflect information discarded by a matching representation and changes in the retained population. The theorem is specific to CRM; the CEM result is an empirical example of the broader distinction.

Table 4: Criteo observational benchmark ($\alpha = 2.0$; $n_T = 30{,}000$; 5 random seeds; mean $\pm$ 95% CI). |Bias|/SE: absolute ATE error divided by the true ATE SE (= 0.00132) from the full 13.98M randomized dataset. Max log-VR: $\max_j \left|\log(\hat{\sigma}^2_{T,j}/\hat{\sigma}^2_{C,j})\right|$ across $j = 1, \ldots, p$ covariates. Bold: best per column. CEM achieves the lowest balance metrics across every column yet the highest estimation error, showing that marginal balance rankings and estimation-error rankings can differ.

| Method | MaxSMD | Mean SMD | Max log-VR | |Bias|/SE | Retention | RT (s) |
|---|---|---|---|---|---|---|
| Naive (unmatched) | 0.818 | — | — | — | 100% | — |
| CRM $k = 1$ | $0.012 \pm 0.001$ | $0.005 \pm 0.001$ | $0.191 \pm 0.082$ | $54.5 \pm 7.3$ | 99.5% | **0.5** |
| CRM-CAM | $0.011 \pm 0.001$ | $0.005 \pm 0.001$ | $0.134 \pm 0.052$ | $54.3 \pm 7.8$ | 99.5% | **0.5** |
| PS-Subclass | $0.015 \pm 0.003$ | $0.006 \pm 0.001$ | $0.187 \pm 0.065$ | $51.2 \pm 8.9$ | 100.0% | 4.8 |
| PSM 1:1 | $0.020 \pm 0.006$ | $0.006 \pm 0.001$ | $0.454 \pm 0.198$ | $\mathbf{50.6} \pm 6.1$ | 100.0% | 7.0 |
| CEM | $\mathbf{0.002} \pm 0.001$ | $\mathbf{0.001} \pm 0.000$ | $\mathbf{0.070} \pm 0.024$ | $66.5 \pm 6.2$ | 91.5% | 21.9 |
| FLAME-lite | $0.014 \pm 0.001$ | $0.004 \pm 0.001$ | $0.065 \pm 0.017$ | $50.8 \pm 6.5$ | 99.9% | 94.9 |

**CRM balance and variance ratio.** At the headline configuration CRM $k = 1$ attains MaxSMD = 0.012, comparable to the other matchers, while retaining 99.5% of treated units. Its Max log-VR ($0.191 \pm 0.082$) is competitive with the collision-free baselines and substantially lower than the naive 1:1 PSM, indicating that CRM controls second-moment balance without the variance inflation that caliper-based exclusion can induce. We caution that the strongest, most robust Criteo differentiator is computational rather than a strict balance win (below): once the PSM baseline is made collision-free, its marginal balance becomes comparable to CRM's (see the verification paragraph), so we frame CRM here as *competitive at scale and far cheaper*, not as uniformly dominant on MaxSMD.

**Verification with the collision-free PSM.** Because the simulation correction (Section 6.2) changed the PSM baseline, we re-ran the Criteo comparison using the collision-free sorted-score matcher over the full scaling grid reported below. On Criteo's good-overlap regime the correction mainly changes the interpretation of retention: sorted PSM retains essentially all treated units, so CRM's advantage is not due to an implementation artifact in the baseline. Instead, the corrected comparison shows that CRM remains competitive on bias and balance while retaining its main design advantage: it does not solve a nearest-neighbor assignment problem. The evaluated sorted-score PSM is nevertheless an efficient scalar matcher, so the runtime gap below should be interpreted as an implementation benchmark rather than a lower bound for all PSM algorithms. Across the full corrected grid (Table 5; $\alpha \in \{0.5, 2, 4\}$, $n_T$ up to 200,000, three seeds, 36 cells), CRM $k = 1$ attains lower MaxSMD than the collision-free PSM in 31 of 36 cells and is faster in all 36, with a median speedup of 8.5×. We read this as a consistent small balance edge combined with a large and uniform computational advantage, not as a claim of strict balance dominance.

Table 5: Criteo corrected grid: CRM $k = 1$ vs. collision-free PSM across $\alpha \in \{0.5, 2, 4\}$, $n_T \in \{10\text{k}, 30\text{k}, 100\text{k}, 200\text{k}\}$, 3 seeds (36 cells). CRM attains lower MaxSMD in 31/36 cells and is faster in all 36, with median speedup 8.5×.

| $\alpha$ | cells | CRM lower MaxSMD | median MaxSMD (CRM / PSM) | median speedup |
|---|---|---|---|---|
| 0.5 | 12 | 9/12 | 0.006 / 0.011 | 8.4× |
| 2.0 | 12 | 12/12 | 0.006 / 0.012 | 9.8× |
| 4.0 | 12 | 10/12 | 0.008 / 0.011 | 8.8× |
| **all** | 36 | **31/36** | – | **8.5×** |

**Computational efficiency.** CRM and CRM-CAM complete in 0.5 s, versus 7.0 s for PSM (14×), 21.9 s for CEM (44×), and 94.9 s for FLAME-lite (190×).

Table 6: Criteo robustness with the *collision-free* sorted-score PSM ($n_T = 30,000$; 3 seeds; means). All PSM comparisons in this corrected grid use `psm_sorted_match`.

| Method | $\alpha = 0.5$ | | | $\alpha = 2.0$ | | | $\alpha = 4.0$ | | |
|---|---|---|---|---|---|---|---|---|---|
| | \|B\|/SE | SMD | Ret. | \|B\|/SE | SMD | Ret. | \|B\|/SE | SMD | Ret. |
| CRM $k = 1$ | 12.1 | 0.009 | 100.0% | 15.7 | 0.007 | 100.0% | 6.8 | 0.009 | 100.0% |
| PSM sorted | 14.2 | 0.017 | 100.0% | 24.3 | 0.016 | 100.0% | 13.7 | 0.018 | 100.0% |

## 7.3 Robustness Across Confounding Strengths and Sample Sizes

We reran the full Criteo scaling grid with the collision-free sorted-score PSM implementation: $n_T \in \{10,000, 30,000, 100,000, 200,000\}$, $n_C = 2,000,000$, $\alpha \in \{0.5, 2.0, 4.0\}$, and three seeds. The corrected run removes the earlier retention artifact: PSM sorted now retains essentially all treated units, matching CRM's near-complete retention. The substantive conclusion is therefore sharper and more conservative: CRM is not advantaged because PSM accidentally drops treated units; instead, CRM reaches comparable or better balance on this grid while using a centroid-referenced computation that is much faster. At the headline $n_T = 30,000$ setting in Table 6, CRM has lower mean MaxSMD at all three confounding levels and lower mean |Bias|/SE than sorted PSM. Figure 9 displays the full corrected scaling curves.

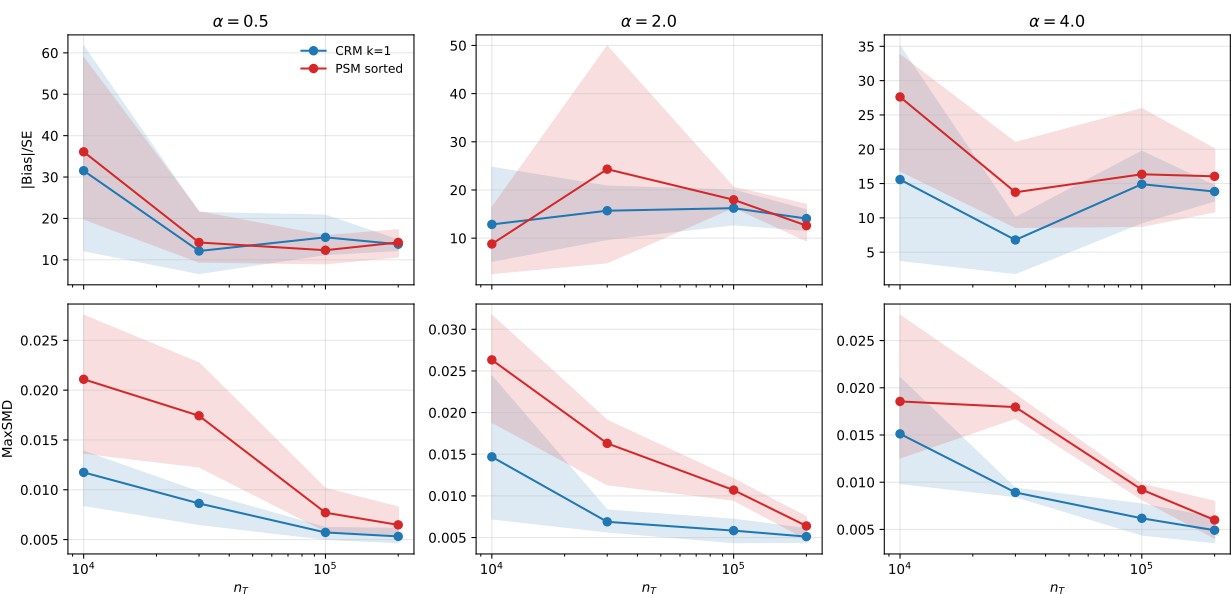

Figure 9: Criteo large-scale benchmark ($n_C = 2,000,000$; 3 seeds; mean $\pm$ 95% CI). **Top row (A–C):** ATE estimation error (|Bias|/SE) vs. $n_T$ at mild ($\alpha = 0.5$), moderate ($\alpha = 2.0$), and strong ($\alpha = 4.0$) confounding. **Bottom row (D–F):** Max SMD vs. $n_T$ at the same three levels using the corrected collision-free sorted-score PSM baseline. CRM and PSM sorted both retain nearly all treated units; CRM's advantage is that it obtains comparable or lower imbalance without pairwise nearest-neighbor search, so the robust differentiator remains runtime (Figure 4).

## 8 NHANES Real-Data Application

### 8.1 Data and Setting

We apply CRM, CRM-CAM, and propensity-score matching to a real public NHANES 2017–2018 extract (Centers for Disease Control and Prevention (CDC), 2020). Treatment is daily cigarette smoking ($n_T = 668$); controls are non-daily smokers and non-smokers ($n_C = 4{,}200$). The outcome is systolic blood pressure (mmHg). The 13 pretreatment covariates are age, sex, BMI, HDL cholesterol, diabetes, hypertension, physical activity, alcohol use, education, income-to-poverty ratio, and three race/ethnicity indicators.

### 8.2 Pre-Matching Diagnostic

The real NHANES extract does not exhibit the sharp alcohol boundary used in the earlier diagnostic illustration. The largest pre-matching imbalance is income-to-poverty ratio (pre-match MaxSMD = 0.599), followed by alcohol use (SMD = 0.430). The shortage fraction is small, $\hat{\pi} = 0.018$, indicating limited support shortage rather than a dominant single-covariate overlap failure. Thus the NHANES analysis is best read as a real-data sensitivity check: it asks whether CRM behaves reasonably on a public health survey with mixed continuous, binary, and categorical covariates, not whether a severe structural positivity violation is present.

### 8.3 Results

Table 7: Real NHANES 2017–2018 smoking–blood-pressure application. ReprSMD is the MaxSMD between the matched treated subsample and the full treated population, measuring sample distortion. The shortage fraction is small ($\hat{\pi} = 0.018$), and no method estimates a positive smoking association with systolic blood pressure on this extract.

| Method | MaxSMD | Retention | ReprSMD | ATT (mmHg) | 95% CI |
|---|---|---|---|---|---|
| CRM standard | 0.210 | 91.3% | 0.071 | -1.44 | [-3.87, 0.53] |
| CRM-CAM | 0.229 | 100.0% | 0.000 | -1.86 | [-3.89, 0.04] |
| PSM 1:1 | 0.085 | 98.7% | 0.054 | -1.01 | [-3.09, 1.07] |

Table 7 reports the application. On this real extract, PSM attains the smallest post-match MaxSMD, while CRM and CRM-CAM retain more of the treated population. The three point estimates are all small and negative, and all confidence intervals include zero. This result is useful precisely because it is less dramatic than the synthetic stress tests: CRM's diagnostic reports only a small support shortage, and the matched estimators agree that the extract does not provide evidence for a positive daily-smoking association with systolic blood pressure after adjustment.

**Survey-weight sensitivity.** Because NHANES is a complex survey, we also recompute ATT and balance using the MEC exam weight `WTMEC2YR`. The weighted results in Table 8 leave the qualitative conclusion unchanged: the weighted ATT shifts by at most 0.62 mmHg across the three methods, and none of the methods yields a positive association. Because the real extract contains only mild support shortage, we also report a NHANES-calibrated semi-synthetic stress test in Section C. That experiment uses the real NHANES covariates as the base but imposes a known overlap violation and known treatment effect; it is a diagnostic validation exercise, not a public-health estimate.

## 9 Minimum Reporting Standard

By Theorem 1 and Corollary 1, $\hat{\pi} > 0$ indicates that the full treated population is not supported in the CRM grid. We therefore recommend that matched observational analyses report six quantities: the shortage fraction $\hat{\pi}$ before matching; final treated retention $|S_M|/n_T$ after capacity constraints; the covariates that define the unsupported subgroup $S^c$; the estimand label ("full ATT" only when $\hat{\pi} = 0$ and retention is 100%,

Table 8: Survey-weight sensitivity on the real NHANES extract. ATT and MaxSMD are recomputed with the MEC exam weight `WTMEC2YR`. The shortage fraction $\hat{\pi} = 0.018$ is support-based and invariant to the weights.

| Method | $\text{ATT}_{\text{unw}}$ | $\text{ATT}_{\text{wt}}$ | $\text{MaxSMD}_{\text{unw}}$ | $\text{MaxSMD}_{\text{wt}}$ | Ret. | Ret.$_{\text{wt}}$ |
|---|---|---|---|---|---|---|
| CRM standard | -1.44 | -1.26 | 0.210 | 0.309 | 91.3% | 89.5% |
| CRM-CAM | -1.86 | -2.49 | 0.229 | 0.303 | 100.0% | 100.0% |
| PSM 1:1 | -1.01 | -0.68 | 0.085 | 0.155 | 98.7% | 98.5% |

otherwise a conditional ATT for the retained overlap population); any auxiliary information about $\tau_{S^c}$; and a plain scope statement describing the treated population to which the estimate applies. When retention is below $1 - \hat{\pi}$, the scope statement must also describe capacity-driven exclusions. In the real NHANES example, $\hat{\pi} = 0.018$, so the support diagnostic excludes at least 1.8% of daily smokers before any additional matching attrition is considered.

## 10 Discussion

The empirical results support a narrow interpretation. CRM is not a replacement for every high-quality matcher. In moderate samples, corrected PSM and entropy balancing often achieve better marginal balance. CRM's strength is the large-sample design regime: it avoids global pairwise search, retains nearly all treated units when overlap is adequate, and reports both the pre-matching shortage fraction and final retention. This is why the Criteo benchmark is the central application, while the simulation study is used mainly to identify the conditions under which CRM is or is not competitive.

Three limitations follow directly from the construction. First, CRM requires representation adequacy: Assumption 4 is not implied by ignorability given $X$, and nonlinear or multi-modal treatment assignment can leave residual confounding in $Z = (d, \phi)$. Second, CRM trades some finite-sample marginal balance for retention and scalability; when the goal is the best possible MaxSMD in a small overlap population, PSM or weighting may be preferable. Third, CRM's bootstrap intervals are approximate calibration tools rather than efficiency guarantees, as discussed below.

CRM is also compatible with outcome-modeling and doubly robust estimation. In supplementary experiments, applying AIPW after CRM matching reduces absolute bias by roughly a factor of four and performs similarly to a correctly specified AIPW-OLS estimator (Figure 14). We therefore view CRM as a design-stage method that can be followed by model-based estimation, not as a competitor to the entire class of doubly robust procedures. Cross-fitted double/debiased machine learning estimators (Chernozhukov et al., 2018) are a natural estimation-stage complement, but a full cross-fitted super-learner benchmark is outside the matching-design comparison in the main paper.

### 10.1 Uncertainty Quantification

CRM reports paired-bootstrap uncertainty intervals as an *approximate* measure of variability (Algorithm 1, Stage 3). The paired bootstrap treats matched pairs as i.i.d. and ignores uncertainty from the learned partition, so we use it as an empirical uncertainty summary rather than a formal post-matching variance estimator.

A calibration study across $n_T \in \{500, 1{,}000, 2{,}000, 5{,}000, 10{,}000\}$ (100 replications each, Scenario SS3) finds sub-nominal coverage of 0.745–0.870 for the paired-bootstrap 95% interval (Figure 10). The interval width shrinks with sample size, but the nominal coverage is not guaranteed. This is consistent with Abadie & Imbens (2008), who show that the nonparametric bootstrap can fail for nonsmooth matching estimators.

To separate variance calibration from estimator bias, Section B adds an inference diagnostic with 500 replications per setting. The diagnostic compares paired-bootstrap standard-error Wald intervals, conservative cell-stratified Wald intervals, and outcome-model bias corrections. In three mean-shift settings the CRM

bias is small, but both bootstrap and cell-stratified standard errors underestimate Monte Carlo variability by roughly 20–25%. In the covariance-driven nonlinear setting, basic CRM has large representation bias; a linear bias correction misses it, while a quadratic correction detects and removes it in that constructed setting. We therefore treat CRM intervals as approximate uncertainty summaries and use the bias-correction gap as an additional diagnostic for residual prognostic imbalance.

**For practitioners, we recommend:**

- Report the paired-bootstrap interval from the software as an approximate uncertainty summary, not as a guaranteed nominal confidence interval.
- When CRM cells are central to the analysis, also report the conservative cell-stratified standard error in Section B; it is a sensitivity calculation conditional on the learned partition.
- Use the bias-correction gap in Section B to audit whether the CRM representation leaves prognostic imbalance. A large gap indicates that the point estimate is sensitive to outcome modeling.
- Do **not** interpret these intervals as semiparametric efficiency-optimal intervals. The $O(n_T^{-1/2})$ rate result (Proposition 4) concerns MSE convergence in fixed representation dimension, not Cramér–Rao efficiency.

Fully valid post-matching inference for CRM remains a theoretical problem: it must account simultaneously for the learned partition, nonsmooth matching, finite-cell sampling, and possible representation bias.

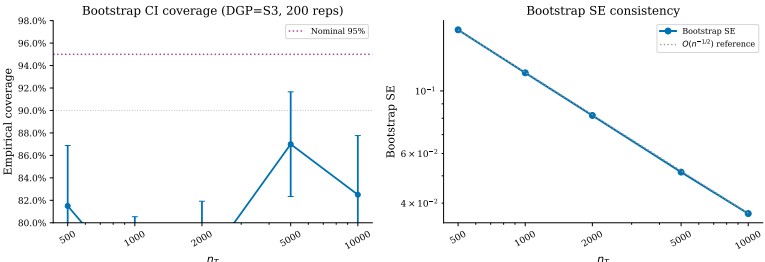

Figure 10: Empirical coverage of the paired bootstrap 95% interval vs. $n_T$ (Scenario SS3; 100 replications). Coverage stays in the 0.745–0.870 band, below nominal, while the interval width (right axis / annotation) shrinks correctly with $n_T$. The gap reflects the nonstandard calibration of nonsmooth matching estimators (Abadie & Imbens, 2008); intervals are reported as empirical calibration, not guaranteed coverage.

**Standard causal assumptions.** CRM inherits the standard assumptions: no unmeasured confounding (Assumption 2), no interference (Assumption 1), and consistency of potential outcomes.

## 10.2 Future Extensions

Several extensions are natural but outside the present contribution. First, the treated mean could be replaced by a robust reference location, such as a trimmed centroid or geometric median, to improve stability under heavy-tailed covariates. A multi-centroid version could similarly use treated clusters or medoids as local reference objects, addressing multimodal treated distributions while preserving CRM's reference-based matching structure. Second, variance-level or interaction-based confounding could be addressed by adding a covariance-direction coordinate to the current mean-shift Fisher coordinate. Third, a fully design-based variance estimator could replace the approximate bootstrap used here. Fourth, a capped CRM-NN implementation could retain nearest-neighbor refinement in small cells while bounding large-scale overhead.

## 10.3 Software and Reproducibility

CRM, all variants (CRM-random, CRM-NN, CRM-RunFast), and the pre-matching diagnostic are implemented in the Python package `crm-matching`. A public repository and archival DOI will be released upon acceptance.

An **anonymized reproducibility supplement** (ZIP, $\leq 100$ MB) is included as supplementary material on OpenReview. It contains all scripts, configuration files, and the `crm` package required to reproduce every figure and table from raw data. The deterministic reproduction path is:

1. `pip install -r requirements.txt`
2. `pytest tests/`                                        (<2 min; all tests pass)
3. `python experiments/run_simulation.py -fast`             (smoke-test; <5 min)
4. `python experiments/run_simulation.py`                  (full; $\approx 2$ h)

All random seeds are passed explicitly; no global state is used. Replication $i$ uses seed `seed_base` $+ 1000i$, matching the paper tables exactly.

## 11 Conclusion

We have introduced Centroid-Referenced Mahalanobis Matching (CRM), a representation-based design for causal inference in large observational studies. By replacing pairwise unit-level matching with stratified distributional matching over a two-dimensional geometric summary $(d, \phi)$, CRM avoids the $O(n_T n_C)$ pairwise-search term; its implemented cost is $O(np^2 + p^3 + n \log n)$, simplifying to $O(np^2 + n \log n)$ when $n \geq p$, and it has an $O(n_T^{-1/2})$ MSE bound in fixed representation dimension.

In three of the four simulation designs, adding the Fisher coordinate substantially reduces the imbalance of radial-only CRM. These experiments support $k = 1$ as the default for the tested mean-shift settings, without claiming that one direction is sufficient under arbitrary assignment. The Criteo benchmark demonstrates two complementary findings: MaxSMD is an insufficient proxy for estimation accuracy (CEM's near-zero MaxSMD coexists with the highest estimation error), and CRM matches or improves on the balance of a correctly implemented (collision-free) PSM across the large-scale configurations while retaining at least 99.4% of treated units and running roughly an order of magnitude faster. The LaLonde application (LaLonde, 1986) illustrates CRM-NN in a small sample, reducing bias from $-\$908$ to $+\$135$ while retaining more treated units than full Mahalanobis NN. The NHANES application provides a real-data check: the pre-matching shortage diagnostic reports only a small support restriction, while survey weighting does not materially change the smoking–blood-pressure estimate.

As observational datasets continue to grow in scale and complexity, methods that jointly address computational scalability, transparent estimand definition, and transparent reporting of overlap limitations will become increasingly important. CRM represents a principled step in this direction.

### Broader Impact Statement

CRM is a methodological contribution to statistical causal inference. Matching methods are widely used in policy evaluation, medical research, and social science. CRM's pre-matching shortage diagnostic explicitly surfaces estimand restrictions that are implicit in all matching methods, promoting more transparent reporting and reducing the risk of overgeneralizing causal findings. The pre-matching diagnostic also helps practitioners identify when additional data collection is needed rather than proceeding with a structurally deficient analysis. No direct negative societal impacts are anticipated from this methodological work, beyond the standard risks that any causal inference method carries when applied to observational data that may contain unmeasured confounding.

### Acknowledgments

Omitted for anonymous review.

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

## A    Balance–Retention Frontier

Figure 11 displays the two components of CRMSE without imposing equal weighting, in the spirit of the balance–sample-size frontier (King et al., 2017). In S1–S3, entropy balancing or collision-free PSM lies closer to the ideal corner than CRM $k = 1$. In S4, no method achieves both adequate balance and high retention. This figure documents the finite-sample tradeoff and should not be read as evidence that CRM dominates competing methods. CRM's empirical case instead rests on its support diagnostic and its large-sample computational behavior.

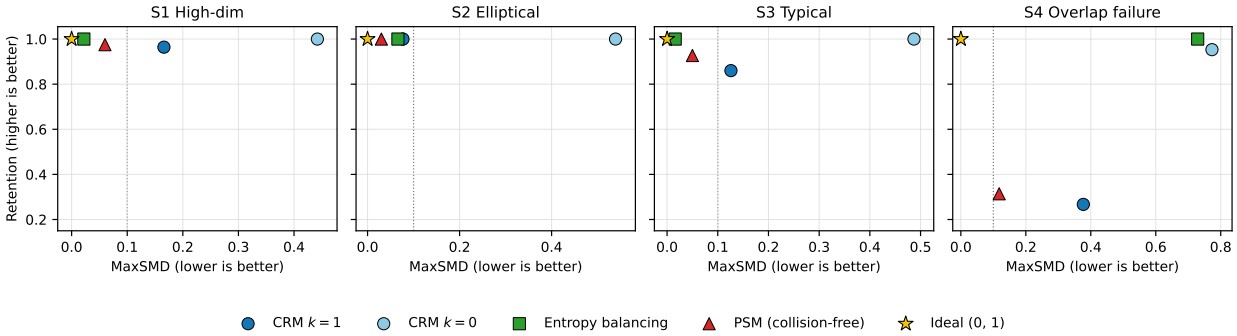

Figure 11: MaxSMD and retention across the four simulation scenarios at $n_T = 1,000$. Each point is a method, and $(0, 1)$ is the ideal of perfect balance and full retention. The dotted line marks MaxSMD $= 0.1$. The figure separates the two components of CRMSE and shows that CRM is not uniformly frontier-dominant in these moderate-sample settings.

## B    Conservative Variance and Bias-Correction Diagnostics

This appendix gives the additional inference diagnostic used in Section 10.1. It is not a new formal inferential guarantee. Its purpose is to separate three mechanisms that can reduce coverage: variance underestimation, bias in the CRM point estimate, and the nonsmooth form of the matching estimator.

**Conservative cell-stratified standard error.** Conditional on the learned CRM partition, treat each occupied cell $k$ as a stratum. Let

$$\hat{\tau}_k = \bar{Y}_{T,k} - \bar{Y}_{C,k}, \qquad w_k = \frac{n_{T,k}}{\sum_{\ell} n_{T,\ell}},$$

where $n_{T,k}$ and $n_{C,k}$ count distinct matched treated and control units in cell $k$. The cell-stratified point estimate is $\sum_k w_k \hat{\tau}_k$, and the conservative variance calculation is

$$\widehat{\mathrm{Var}}_{\mathrm{cell}}(\hat{\tau}) = \sum_k w_k^2 \left( \frac{s_{T,k}^2}{n_{T,k}} + \frac{s_{C,k}^2}{n_{C,k}} \right).$$

For sparse cells with $n_{T,k} < 2$ or $n_{C,k} < 2$, the undefined within-cell variance is replaced by the pooled within-cell variance from non-sparse cells on the same side. This mirrors variance calculations for subclassification and coarsened exact matching (Rosenbaum & Rubin, 1984; Iacus et al., 2012), but here it is used only as a conservative sensitivity calculation conditional on CRM's learned cells.

**Outcome-model bias correction.** To diagnose residual prognostic imbalance, fit a control-outcome model $\hat{\mu}_0(x)$ on the control group, following the bias-corrected matching logic of Abadie & Imbens (2011). We write BC for bias correction and compute

$$\hat{\tau}_{\text{BC}} = \hat{\tau}_{\text{CRM}} - \sum_k w_k \left\{ \bar{\hat{\mu}}_{0,T,k} - \bar{\hat{\mu}}_{0,C,k} \right\}.$$

The correction term is a bias audit: a large value means that treated and matched control units still differ in predicted untreated outcomes. We report both a linear correction and a quadratic correction. The quadratic correction is not part of the basic CRM matching design; it is included to test whether a flexible outcome model can detect covariance-driven nonlinear confounding.

**Coverage diagnostic.** We run 500 replications in four inference-diagnostic settings, denoted I1–I4 to avoid confusion with the main simulation labels: I1 is a linear high-dimensional mean-shift setting, I2 is a correlated elliptical mean-shift setting, I3 is a covariance-driven nonlinear failure case, and I4 is a multi-axis mean-shift setting. All rows in Table 9 use Wald intervals so that the bootstrap and conservative standard-error estimates are compared on the same scale. The software also reports percentile paired-bootstrap intervals; neither form is claimed to have guaranteed nominal coverage.

Table 9: Inference diagnostic coverage at nominal 95% level (500 replications per setting). "Bootstrap-SE" means a Wald interval using the paired-bootstrap standard error. "Conservative" uses $\widehat{\text{Var}}_{\text{cell}}$ above. The empirical-bias-corrected row is simulation-only and subtracts the Monte Carlo mean bias before checking coverage.

| Interval | I1 | I2 | I3 | I4 |
|---|---|---|---|---|
| Paired bootstrap-SE Wald | 0.886 | 0.900 | 0.000 | 0.906 |
| Cell-stratified conservative Wald | 0.884 | 0.898 | 0.000 | 0.906 |
| Linear bias-corrected conservative Wald | 0.994 | 0.988 | 0.000 | 0.992 |
| Quadratic bias-corrected conservative Wald | 0.994 | 0.990 | 1.000 | 0.990 |
| Empirical-bias-corrected diagnostic | 0.888 | 0.898 | 0.702 | 0.900 |

Table 9 shows two different failure modes. In I1, I2, and I4, CRM's average bias is close to zero, but the bootstrap and conservative SEs are about 0.74–0.82 of the Monte Carlo SD. The problem is therefore mainly variance calibration rather than estimator centering. In I3, the raw CRM bias is large ($-1.713$), the linear bias correction does not detect it, and no variance estimator can restore coverage. The quadratic correction detects almost exactly the missing prognostic structure (mean correction 1.712) and restores coverage in this constructed setting. Thus, the bias-correction gap is useful as a diagnostic for representation failure, but formal valid inference for CRM remains future work.

**NHANES comparison.** For the real NHANES smoking application ($n_T = 668$), we report paired bootstrap intervals as approximate empirical uncertainty summaries. Because NHANES is a complex survey, Table 8 separately reports survey-weighted point estimates and weighted balance; a full design-based variance estimator with strata and primary sampling units is outside the scope of this methodological comparison.

## C  NHANES-Calibrated Semi-Synthetic Stress Test

The corrected real NHANES extract has only mild support shortage ($\hat{\pi} = 0.018$), so it is not a stress test for the estimand-decomposition diagnostic. We therefore add a semi-synthetic experiment using the same real NHANES covariate extract as the base, while synthetically imposing an alcohol-support barrier and a known heterogeneous treatment effect. This experiment is not used as evidence about smoking and blood pressure; its purpose is to check whether the CRM diagnostic and conditional-estimand interpretation behave as expected when the support problem is known. Table 10 reports the result.

Table 10: NHANES-calibrated semi-synthetic overlap stress test. The real NHANES covariate extract is used as the base, but alcohol support and the outcome are synthetically generated with known truth. The imposed shortage fraction is $\hat{\pi} = 0.049$; the true full ATT is 7.65 mmHg and the true supported ATT is 7.45 mmHg. $\text{Bias}_{\text{matched}}$ is computed relative to the true ATT of each method's retained treated subset.

| Method | Ret. | MaxSMD | ReprSMD | ATT | True $\text{ATT}_{\text{matched}}$ | $\text{Bias}_{\text{matched}}$ |
|---|---|---|---|---|---|---|
| CRM standard | 89.4% | 0.266 | 0.174 | 7.61 | 7.53 | 0.09 |
| CRM-CAM | 99.9% | 0.428 | 0.006 | 7.49 | 7.64 | -0.15 |
| PSM 1:1 | 94.6% | 0.056 | 0.192 | 7.23 | 7.49 | -0.26 |

# D Proofs

## D.1 Proof of Proposition 4 (Conservative Rate Bound)

We follow the standard bias-variance decomposition for histogram regression estimators in $d_Z = 2$ dimensions (Devroye & Györfi, 1985).

**Bias.** Within cell $k$ with center $z_k$ and diameter $h_n$, the cell control mean $\hat{m}_k = |C(k)|^{-1} \sum_{j \in C(k)} Y_j(0)$ approximates $\mathbb{E}[Y(0) \mid Z = z_k, T = 0]$. Under Lipschitz continuity of $m(z) = \mathbb{E}[Y(0) \mid Z = z]$ with constant $L$, the within-cell approximation satisfies $\left| \mathbb{E}[\hat{m}_k] - m(z_k) \right| \leq L h_n + |\Delta(z_k)|$. Under Assumption 4, $\Delta \equiv 0$, giving $\text{Bias}^2 = O(h_n^2)$.

**Variance.** Each cell contains on average $n_C h_n^{d_z} f_Z(z_k)$ control units, where $f_Z$ is the density of $Z(X)$. Hence $\text{Var}(\hat{m}_k) = O(1/(n_C h_n^{d_z}))$. The ATT estimator averages $\hat{\tau}_{\text{CRM}} = |S|^{-1} \sum_{i \in S}(Y_i - \hat{m}_{k(i)})$. The treated outcomes $Y_i(1)$ contribute $O(1/n_T)$ to variance, and bounding the variance of the weighted average by the largest cell-mean variance gives the conservative order $O(1/(n_T h_n^{d_z}))$ when $n_C \propto n_T$. This bound does not exploit the additional variance reduction from averaging across cells and is therefore not asserted to be sharp.

**Balancing the bound.** Setting $\text{Bias}^2 = \text{Variance}$: $h_n^2 = 1/(n_T h_n^{d_z})$, so $h_n^{2+d_z} = 1/n_T$, giving

$$h_n^* = n_T^{-1/(2+d_z)}.$$

Substituting into $\text{MSE} = O(h_n^{*2})$:

$$\text{MSE}(\hat{\tau}_{\text{CRM}}) = O\left(n_T^{-2/(2+d_z)}\right).$$

For $d_Z = 2$: $\text{MSE} = O(n_T^{-1/2})$. For $d_Z = p$ (full-dimensional matching): $\text{MSE} = O(n_T^{-2/(2+p)})$. The ratio is $n_T^{(p-2)/(2(2+p))}$, which diverges for $p > 2$. □

# E PCA-Matching Sensitivity

Figure 12 reports the sensitivity of PCA-95%+NN balance to the variance-retention threshold and the number of retained components, the comparison referenced in Section 6.4. Unlike CRM's default two-coordinate representation, PCA-matching requires a variance-retention threshold and its balance varies non-monotonically with that choice: retaining too few components discards directions relevant to treatment or outcome, while retaining too many reintroduces the high-dimensional matching problem CRM is designed to avoid.

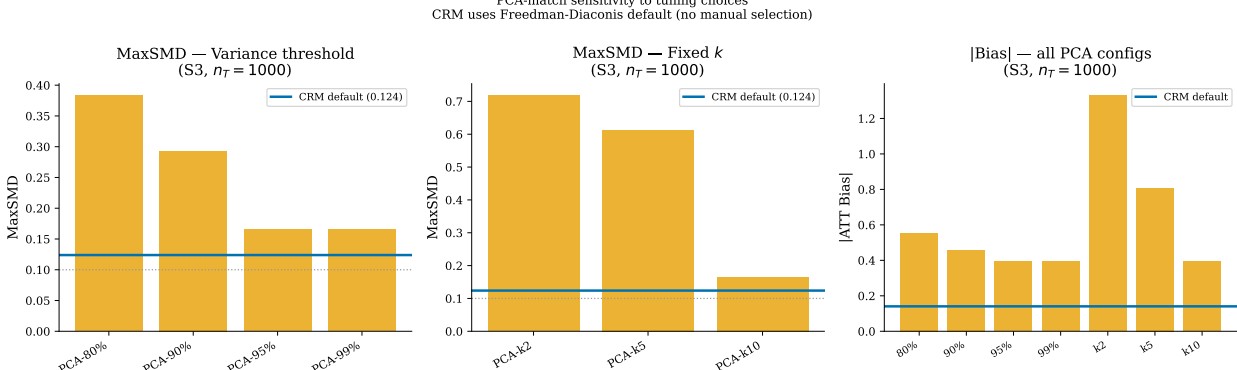

Figure 12: Balance of PCA-95%+NN as a function of the retained variance fraction / number of principal components. MaxSMD varies non-monotonically with the threshold, with no stable "correct" setting across data-generating processes; CRM has no analogous tuning knob.

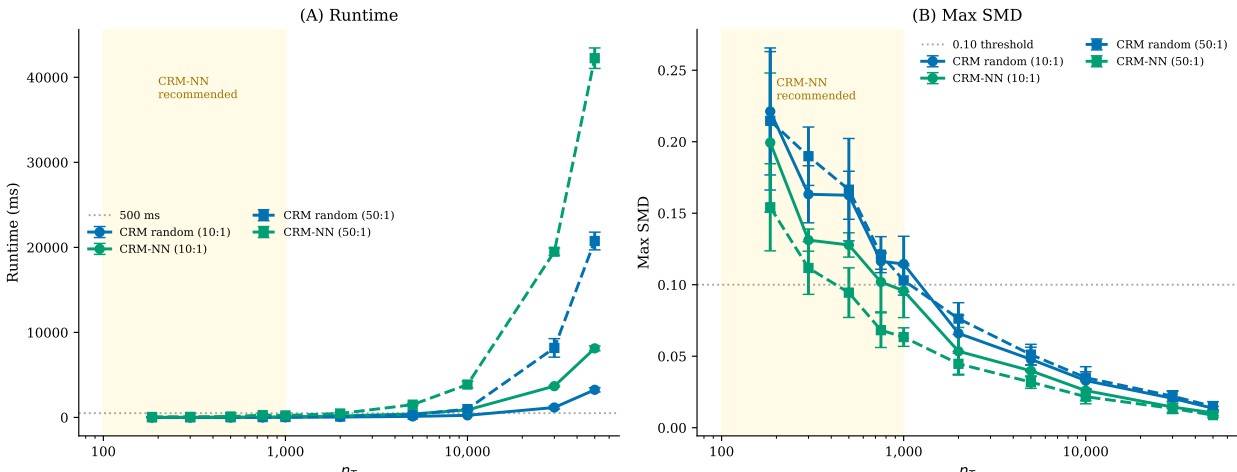

Figure 13: CRM-random vs. CRM-NN across sample sizes ($p = 8$, Scenario A, 5 data seeds $\times$ 7 timing replicates; gold shading marks the CRM-NN recommended regime $n_T < 1{,}000$). **(A)** Runtime: NN overhead stays below 500 ms for $n_T \leq 1{,}000$ at both tested control ratios, then grows proportionally to $n_C/n_T$. **(B)** Max SMD: the quality gap persists at all $n_T$ because average controls per cell grows as $\bar{n}_C^{\text{cell}} \propto n_T^{1/3}$, maintaining a pool within each cell for NN to improve upon.

## F  CRM-NN Timing Supplement

## G  Decision Flowchart

---

**Method Selection Flowchart**

**Step 1. Sample size.**

- $n_T < 200 \rightarrow$ Use 1:1 Mahalanobis NN (cells too sparse for binning).
- $200 \leq n_T < 1{,}000 \rightarrow$ Use CRM-NN (Section 4.4).
- $n_T \geq 1{,}000 \rightarrow$ Use CRM-random (Section 4.3).

**Step 2. Shortage fraction.**
Compute $\hat{\pi}$ via Equation (8) (before any matching).

- $\hat{\pi} < 5\% \rightarrow$ Full ATT approximately estimable.
- $5\% \leq \hat{\pi} < 30\% \rightarrow$ Conditional ATT; report scope statement.
- $\hat{\pi} \geq 30\% \rightarrow$ Severe overlap failure; expand control pool or restrict estimand explicitly before proceeding.

**Step 3. Confounding structure.**

- Mixed continuous/binary covariates $\rightarrow$ CRM-CAM (Section 4.6).
- Two independent confounding axes suspected $\rightarrow$ CRM $k = 2$ (future work).
- Default $\rightarrow$ CRM $k = 1$ (Section 4.1).

---

## H  CRM Variants: Full Descriptions

**CRM-CAM (Correlation-Adjusted Mahalanobis).**  Replaces $\hat{\Sigma}_T$ with the correlation-scale matrix $R = D^{-1/2}\hat{\Sigma}_T D^{-1/2} + \varepsilon I_p$, where $D = \mathrm{diag}(\hat{\Sigma}_T)$, placing all covariates on a unit-variance scale. Recommended for mixed continuous/binary covariate sets where raw variance differences would otherwise dominate the Mahalanobis geometry.

**CRM-Pool (Pooled Bin Edges).**  Uses bin edges from the pooled $(d, \phi)$ distribution of both groups, reducing treated attrition when the centroid shift is large and treated units concentrate near the edge of the control support.

**CRM-Trim (Explicit Support Trimming).**  Formally discards treated units with $d(X_i)$ outside the central 90% of the control distance distribution before matching, and reports them separately as an explicit overlap restriction with the associated estimand caveat.

## I  Geometric Supporting Results

**Proposition 5** (Chi distribution of $d$). *Let $X \mid T = 1 \sim \mathcal{N}(\mu_T, \Sigma_T)$ with $\Sigma_T$ positive definite. Then $d^2(X) \mid T = 1 \sim \chi_p^2$ with $\mathbb{E}[d^2] = p$, $\mathrm{Var}(d^2) = 2p$, and $\mathbb{E}[d(X) \mid T = 1] = \sqrt{2}\,\Gamma((p+1)/2)/\Gamma(p/2)$.*

*Proof.* $Z = \hat{L}^{-1}(X - \mu_T) \sim \mathcal{N}(0, I_p)$, so $d^2(X) = \|Z\|_2^2 \sim \chi_p^2$. $\qquad\square$

**Proposition 6** (Independence of distance and direction). *Under $X \sim \mathcal{N}(\mu_T, \Sigma_T)$, $d(X)$ and $u(X) = \hat{L}^{-1}(X - \mu_T)/d(X)$ are statistically independent, with $u(X)$ uniform on $\mathcal{S}^{p-1}$.*

*Proof.* Writing $Z = d \cdot u$ in polar coordinates, the Jacobian is $d^{p-1}$, giving $f_{d,u}(r, \omega) = (2\pi)^{-p/2}\exp(-r^2/2) \cdot r^{p-1}$, which factors as $f_d(r) \cdot f_u(\omega)$. $\qquad\square$

**Proposition 7** (Complexity). *Algorithm 1 has implemented time complexity $O(np^2 + p^3 + n \log n)$ and space complexity $O(p^2 + n)$, where $n = n_T + n_C$. The term $O(np^2)$ is the dominant linear-algebra cost; the $O(n \log n)$ term comes from quantile binning and sorting. The bound contains no $n_T n_C$ pairwise-search term. Brute-force nearest-neighbor matching in p dimensions requires $O(n_T n_C p)$ distance work, although scalar propensity-score matching can use sorting and need not incur this bound.*

*Proof.* Dominant steps: covariance estimation costs $O(n_T p^2)$, Cholesky factorization costs $O(p^3)$, and computing distances and projections for all units costs $O(np^2)$. Sorting or quantile computation for binning costs $O(n \log n)$ in the standard implementation. Thus total implemented time is $O(np^2 + p^3 + n \log n)$ and space is $O(p^2 + n)$. For the regimes considered here, $n \gg p$, so $p^3$ is absorbed by $np^2$. □

## J  Relationship to Doubly Robust Estimation

Doubly robust (DR) estimators such as augmented inverse probability weighting (AIPW), targeted learning, and cross-fitted double/debiased machine learning (Bang & Robins, 2005; Chernozhukov et al., 2018) address a different, estimation-stage question from matching design. They can target full or restricted populations depending on the overlap and weighting choices. The main experiments therefore compare matching designs and report balance and retention, while this appendix provides a focused IPW/AIPW sensitivity analysis based on estimation error. It is not an exhaustive comparison of outcome learners or cross-fitting schemes. Entropy balancing (Hainmueller, 2012) remains in the main simulation as a scalable representative of balancing weights. Figure 14 reports the resulting sensitivity comparison.

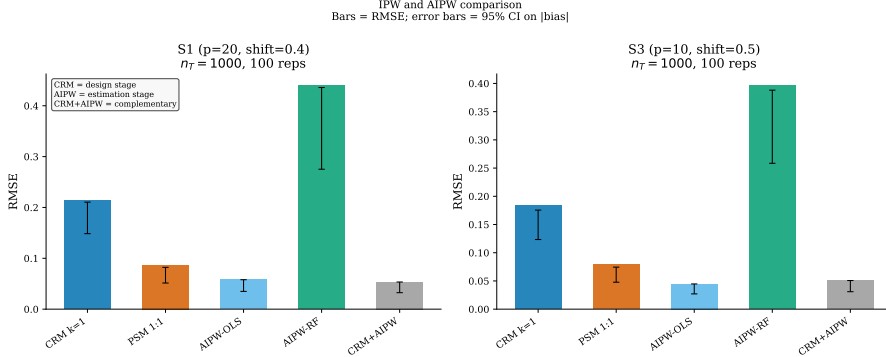

Figure 14: Supplementary comparison with inverse probability weighting (IPW), AIPW, and matching-plus-AIPW estimators. The final estimator can incorporate CRM as a design-stage restriction before AIPW-style estimation.

## K  When CRM May Underperform

1. **High balance requirement at moderate $n_T$.** In the moderate-size synthetic sweeps, PSM often achieves lower MaxSMD than CRM (gap 0.006–0.133 in the reported comparison). If per-covariate MaxSMD below 0.05 is the primary criterion and retention loss is acceptable, PSM is preferable.

2. **Nonlinear confounding.** The linear Fisher direction $v$ cannot capture interaction-based or higher-order confounding. PSM with a flexible treatment model or MALTS (Parikh et al., 2022) may achieve lower bias in such settings.

3. **Very small $n_T$ (below 200).** Even CRM-NN is limited when the average cell contains fewer than 2–3 treated units. Full 1:1 Mahalanobis nearest-neighbor matching is preferable in this regime.

