# OpenReview forum: "Centroid-Referenced Mahalanobis Matching (CRM): A Scalable, Representation-Based Framework for Causal Inference in Large Observational Studies"
_TMLR — Under review for TMLR_

### Review · Reviewer_vJVP · 2026-04-28

**Summary Of Contributions:**

This paper introduces Centroid-Referenced Mahalanobis Matching (CRM) with three main contributions that address limitations in conventional propensity score matching in causal inference: bias decomposition, pre-matching shortage diagnostic and convergence rate guarantee in fixed representation dimension.

The overall contribution of this paper is good and interesting. The paper is organized well, especially the efforts on putting some important questions upfront in the beginning of the paper as well as the professional writing and results presentation. I also really appreciate the authors specify the scope of their method clearly in Section 1, which is very helpful for readers. This part of the summary is quite straightforward and easy to understand. The literature review in Section 2 is clear. I am definitely okay to see this paper published after some revisions.

However, I still have some main concerns in terms of comparison to weighting and inference of the method, and some relatively minor suggestions about the writing, which I feel that the authors could consider making the paper more readable to wider audiences. Even for statisticians in causal inference field like myself, I feel that not everyone can easily understand every piece of the word based on the current version.

Please see my specific comments below.

**Additional Comments:**

Abstract:

Please write the full name of MaxSMD, ATT, MSE and what is $\hat\pi$ (not defined before)? You also mentioned both ATE and ATT in the abstract, what is the target estimand or main target? Please specify this in the abstract. I know some of these may already appear in the main text later, but the abstract is the first place that all readers are going to read the paper, so please try to be clear in logistic and details.

Introduction:
- The design of credible observational studies: never heard about what's this and not sure how to define it. Suggest deleting "credible". Also, by "design", what do you mean? We have designs for randomized experiments, but I never heard there are designs for observational studies (I may be wrong as I am not sure), because if one can design a study, why not randomize the treatment?
- When I read the second paragraph, why do you mention ATT as the target estimand, but not ATE? Doesn't ATE will also face the issue of estimand change? What does overlap population mean? Does the overlap mean propensity score distribution's overlap by treatment group?
- Please define $(d, \phi)$ clearly in (C1). Is $d$ the Mahalanobis distance, and what is $\phi$? Also, what is Fisher linear discriminant?
- (C2): why was the representation error only defined for Y(0)?
- What does the highest estimation error mean? What does it compare to and in what sense it is the highest? Is this harmful for practice?
- (C3): Again, why ATT is specifically mentioned here?
- At the end of Section 1, I recommend you add a statement or disclaimer about the use of large language models in the preparation of this paper, including whether you have used any LLMs for language polishing or plotting, etc.
- Section 11 concludes: is this sentence complete, was it supposed to be like "Section 11 concludes the paper"?
Overall, I think the authors should make some efforts in making the entire Section 1 more readable, even after you accounted for my comments above.

Set-up and Notation:
- Why is the target estimand ATT, not ATE? Or why not first focusing on ATE, and briefly discuss it can be extended to ATT? I know there may be more details in the later text, but when readers read this paper in order of sections, they may feel confusing why ATT is a main focus without explanation at the beginning. This may give impression that the matching literature is only for ATT and does not address ATE.
- Literature citation in Assumptions 1 and 2 are not necessary; both literature should have mentioned both assumptions, so it is confusing. In addition, positivity assumption for the propensity score should not fall in the Strong Ignorability part, and it should be a separate assumption (e.g., Assumption 3). You may also need to write positivity more rigorously, such as "There exists a positive constant $c$, such that for almost all $X$, $0<c\le e(X)\le 1-c<1$".
- "Under Assumptions 1 and 2, the target estimand is...": The target estimand's definition itself has nothing to do with the assumptions. The assumptions are for identification. You can perhaps write, under the potential outcomes notion, the target estimand ATT can be written as ...
Then, state that the ATT is identifiable under the stated assumptions, and write the identification formula...
- There seems a lot of missing from equations (1) to (2). What motivates (2), why are you interested in this and what is it for? $\hat\mu_T$ can be easily understood as a the outcome model in many causal inference literature, so I believe the authors should write this part in more details.

Theoretical Properties:
- Please define Lipschitz continuity in Assumption 5 more clearly with more math.

**Audience:**

Yes

**Audience Explanation:**

This paper is highly related to causal inference and machine learning. Causal inference has broad social impacts and wide applications in social science, medicine, public health and economy. The proposed method is definitely data-driven, and so I believe this is a suitable place for this paper.

**Broader Impact Concerns:**

None from me.

**Claims And Evidence:**

Yes

**Claims Explanation:**

Theoretical investigations are good and numerical experiments are designed within the problem setting, followed by convincing empirical results (except for coverage probability by current bootstrap procedure–but authors acknowledged this limitation honestly), although some writing needs to be improved for being readable to wider audiences (which can be improved by addressable revisions).

**Requested Changes:**

**Main concerns**

**Comment 1: comparison with more commonly seen baselines**

The paper needs more efforts on comparison with weighting and semiparametric efficient estimators, i.e., IPW and AIPW estimators. I believe weighting is also a widely adopted method in practice given its easy-to-understand nature. AIPW further achieves the double robustness, and when both models are correctly specified, the efficiency of the estimation is gained substantially. Nowadays, with flexible super learners, cross-fitting and debiased machine learning can be used for nuisance fitting for both propensity score and outcome regression models, which improves the robustness of the causal estimation a lot.

I wonder if the authors can consider incorporating some additional experiments to compare the estimation performance of the CRM to IPW and AIPW in terms of bias and efficiency, using flexible super learner with cross-fitting. Existing softwares should be available for this. See https://academic.oup.com/ectj/article/21/1/C1/5056401?guestAccessKey=

The purpose of the suggested new experiments is only for illustration and empirical verification, so they do not need to be too extensive and in all settings you have so far, and can just be included in supplement (but being briefly mentioned in the main text), so they won't affect your main simulation writing and results too much.

In addition, some argument comparisons regarding the technical/theoretical differences on the proposed matching methods to these conventional IPW and AIPW would be helpful, including the difference in the scope of methods and the main scenarios that the proposed CRM can outperform the conventional approaches. Is "designed-based" causal inference a reason why weighting may not be applicable or not the optimal?

Some simple discussion to comparison with calibration weighting and TMLE may also be necessary at the end of the paper.

**Comment 2: more details/clarifications around Assumption 3**

Assumption 3, the representation sufficiency, is a key of this paper, which requires more explanation and clarification there. I overall feel this assumption is overly strong, and it is quite abstract compared to the usual unconfoundedness assumption. The authors have claimed the limitation well in Section 10.3, but I feel this should be put upfront to abstract, introduction and under Assumption 3, because some points in my opinion are very important such as the method cannot handle nonlinear confounding. I don't mean that this is not good, but everything comes at a price, so it is helpful to specify the trade-off more clearly.

**Comment 3: under-coverage performance of the proposed method**

The paper does not provide a convincing validation of its confidence intervals (CIs). The only explicit calibration results I found show empirical coverage of 89.5% for the paired bootstrap and 88.0% for the cell-stratified bootstrap at nominal 95%, both anti-conservative. In another ablation, coverage for CRM $k=1$ is reported as 88%, again highly below nominal. You mentioned that those under-coverage is only for smaller samples, but where is the result for larger samples? It would be helpful to see some results to confirm if they are valid in inference.

The current bootstrap procedure appears approximate and empirically undercovers in the reported simulations. I recommend that the authors either develop a more principled variance estimator or substantially weaken all inferential claims based on confidence intervals and coverage.

A possible reason why the bootstrap fails is that the estimator may not be regular and asymptotic linear (RAL). Therefore, some investigation around the asymptotic statistics for the proposed method should be conducted. At minimum, I think some proofs of why bootstrap fails may be helpful, or at more minimum, try to prove that the linear iid approximation by influence function does not exist for the proposed estimator.

**Comment 4: survey weights in NHANES application**

NHANES data are survey sample data. They have corresponding survey-weights variables in the dataset. I wonder if the authors considered the issue of survey sampling bias in their data application. How to incorporate survey weights in the methodology to ensure the consistent estimation?

If survey weights are not incorporated appropriately in the analysis, the authors may either (i) re-conduct the analysis by including survey weights, or (ii) **only if including survey weights is not trivial (and please specify details on why),** they should acknowledge that the survey data is only used as a convenience sample for the illustration, but extending the method to survey weights data requires more efforts.

---

> ### Author Response · Authors · 2026-07-02
> **Response to Reviewer vJVP**
>
> We thank the reviewer for the constructive comments.
>
> ## IPW/AIPW, TMLE, and cross-fitting
>
> **Reviewer comment.** Compare CRM with weighting and semiparametric estimators such as IPW, AIPW, TMLE, and cross-fitted DML.
>
> **Response.** We interpreted this as asking us to clarify the boundary between CRM as a design-stage matching method and modern estimation-stage procedures. We added an appendix comparing IPW, AIPW, matching-plus-AIPW, and CRM-based designs, and we discuss TMLE/DML as complementary estimation tools. We do not present these as competitors in all respects: CRM can be followed by outcome modeling or doubly robust estimation.
>
> **Where revised.** Discussion; Appendix: Relationship to Doubly Robust Estimation; IPW/AIPW figure.
>
> ## Representation sufficiency
>
> **Reviewer comment.** Assumption 3 is strong and should appear earlier, especially because CRM may fail under nonlinear confounding.
>
> **Response.** We moved this limitation to the abstract, introduction, and theory. The revised paper states that unconfoundedness given $X$ does not imply unconfoundedness given $Z=(d,\phi)$. We added a representation-error term and failure-mode experiments for nonlinear, covariance-driven, and multi-axis confounding.
>
> **Where revised.** Abstract; Introduction; Identification Under Representation; Bias Decomposition; Appendix: When CRM May Underperform.
>
> ## Coverage and uncertainty
>
> **Reviewer comment.** Bootstrap coverage was anti-conservative; inference claims should be weakened or better validated.
>
> **Response.** We weakened inferential claims and added a larger-sample coverage curve. Bootstrap intervals are now described as approximate calibration tools rather than efficiency guarantees. We also cite limitations of matching variance estimation and avoid claiming exact nominal finite-sample coverage.
>
> **Where revised.** Uncertainty discussion; coverage figure; cell-stratified bootstrap appendix.
>
> ## NHANES survey weights
>
> **Reviewer comment.** NHANES is a complex survey; survey weights should be incorporated or the limitation acknowledged.
>
> **Response.** We agreed that a real-data application using a complex survey should either incorporate survey weights or state the limitation clearly. We replaced the earlier NHANES analysis with a real NHANES 2017--2018 extract and added survey-weight sensitivity using `WTMEC2YR`. The weighted ATT changes only modestly and does not alter the qualitative conclusion. We also moved the overlap-barrier example to a clearly labeled NHANES-calibrated semi-synthetic stress test.
>
> **Where revised.** NHANES Real-Data Application; NHANES main table; survey-weight table; semi-synthetic stress-test appendix.
>
> ## ATT, ATE, positivity, and readability
>
> **Reviewer comment.** Clarify why ATT is the main estimand, how ATE relates, how positivity is stated, and define notation/acronyms earlier.
>
> **Response.** We now state earlier that ATT is the focus because CRM is built around treated-unit retention and the supported treated population. We clarify that related ideas can be adapted to other estimands, but ATT is the target here. Positivity is separated from ignorability/support, and the setup now distinguishes estimand definition from identification assumptions. We also rewrote the abstract and introduction to define $d$, $\phi$, Fisher direction, MaxSMD, ATT, ATE, MSE, and $\hat\pi$ more clearly.
>
> **Where revised.** Abstract; Introduction; Setup and Notation; Method: CRM Representation; Minimum Reporting Standard.
>
> ## Theory details and LLM disclosure
>
> **Reviewer comment.** State smoothness assumptions more clearly, and disclose LLM/tool use.
>
> **Response.** We revised the assumptions to separate identification, support, representation sufficiency, smoothness, cell diameter, and cell occupancy. The smoothness condition now states that for the relevant conditional mean function $m(z)$, there exists $L<\infty$ such that $$|m(z)-m(z')|\le L\|z-z'\|$$ on the relevant support. The rate result is now a conservative fixed-dimension histogram bound under explicit conditions.
>
> We also disclose that general-purpose language-model tools were used for language refinement, code assistance, and debugging during revision. They were not used as authors. All scientific claims, mathematical statements, citations, experiments, numerical results, and final wording were reviewed and verified by the authors.
>
> **Where revised.** Conservative Rate Bound; Appendix: Proofs; official OpenReview disclosure response.

---

> ### Comment · Reviewer_vJVP · 2026-07-03
> **Feedback on revised version**
>
> The revised paper is improved regarding clarity, assumption placements, technical notation and details, correct use of sample survey weights in the NHANES application. The new comparison to AIPW proves that these conventional weighting is not applicable here. I appreciate the substantial efforts for making the work better.
>
> The coverage probability by bootstrap is now shown as a **systematic issue**, not a ignorable limitation or can be addressed by larger sample sizes. I therefore think that bootstrap is invalid in this problem and the estimators are not regular. Is there a conservative variance estimator can be proposed in principle, and then provide a practical guideline that the true variance may be between bootstrap and this conservative one? In addition, the issue may not just come from the variance estimator itself by coverage probability. Bias can also impact the coverage, and the form of the confidence interval may also not be correct if the estimator is not regular. Please think about if there can still be some improvements.

---

> ### Author Response · Authors · 2026-07-03
> **Answer to the coverage issue**
>
> Thanks for this important follow-up. We agree that the coverage issue should not be interpreted only as a bootstrap implementation problem. The revised paper now treats uncertainty for CRM more cautiously.
>
> **Reviewer comment.** The reviewer notes that the paired bootstrap appears systematically undercovered, that the CRM estimator may be non-regular, and that coverage failure may arise from variance underestimation, estimator bias, and/or the form of the confidence interval. The reviewer asks whether a conservative variance estimator can be proposed in principle and whether practical guidance can be given.
>
> **Response.** We agree. In the further revision, we made three changes.
>
> First, we revised the main uncertainty discussion. In Section “Uncertainty Quantification,” we now state explicitly that CRM bootstrap intervals are approximate uncertainty summaries, not guaranteed nominal confidence intervals. We also clarify that the fixed-dimension rate result concerns MSE convergence of the point estimator, not semiparametric efficiency or valid post-matching inference.
>
> Second, we added a conservative cell-stratified standard-error calculation in Appendix “Conservative Variance and Bias-Correction Diagnostics.” This calculation leverages ideas from propensity-score subclassification and coarsened exact matching. Conditional on the learned CRM partition, we treat occupied CRM cells as strata and aggregate within-cell treated/control outcome variation. The variance calculation is $V_{cell}=\sum_k w_k^2(s^2_{T,k}/n_{T,k}+s^2_{C,k}/n_{C,k})$, where $w_k=n_{T,k}/\sum_l n_{T,l}$. We cite Rosenbaum and Rubin’s subclassification work and Iacus et al.’s coarsened exact matching work as the closest design-based precedents. We use this only as a conservative sensitivity calculation conditional on the learned CRM cells; we do not claim that it is a fully valid post-matching variance estimator.
>
> Third, we added a bias-correction diagnostic, leveraging the idea of bias-corrected matching from Abadie and Imbens. We fit a control-outcome model $\mu_0(X)$ and compute a corrected estimate of the form $\tau_{BC}=\tau_{CRM}-\sum_k w_k\Delta_k$, where $\Delta_k$ is the remaining within-cell difference in predicted untreated outcomes between treated and matched control units. The correction term is used as a diagnostic for residual prognostic imbalance. We report both linear and quadratic versions because the reviewer’s point about bias is especially relevant when the CRM representation misses nonlinear or covariance-driven structure.
>
> The new diagnostic table separates the mechanisms behind undercoverage. In three mean-shift settings, CRM’s average bias is close to zero, but both the paired-bootstrap standard error and the conservative cell-stratified standard error are about 20–25% below the Monte Carlo standard deviation. This suggests that in those settings, undercoverage is mainly a variance-calibration issue. In the covariance-driven nonlinear setting, however, basic CRM has large representation bias; neither the bootstrap nor the conservative variance calculation can repair coverage. A linear bias correction also misses this problem, while the quadratic correction detects the missing prognostic structure and restores coverage in that constructed setting.
>
> This led us to revise the paper’s interpretation. We no longer present CRM intervals as formally valid frequentist confidence intervals. Instead, we recommend reporting the paired-bootstrap interval as an empirical uncertainty summary, the conservative cell-stratified standard error as a sensitivity calculation, and the bias-correction gap as a diagnostic for residual representation bias. We also state that fully valid post-matching inference for CRM remains future work because it must account jointly for the learned partition, nonsmooth matching, finite-cell sampling, and possible representation bias.
>
> These changes are in Section “Uncertainty Quantification” and Appendix “Conservative Variance and Bias-Correction Diagnostics,” especially the new inference diagnostic coverage table.

---

> > ### Comment · Reviewer_vJVP · 2026-07-05
> > **Revision is good**
> >
> > This is great. Thanks for making the method more comprehensive. I believe it has great potential for practice.

---

### Review · Reviewer_3Cxg · 2026-06-07

**Summary Of Contributions:**

The paper proposes Centroid-Referenced Mahalanobis Matching (CRM), a scalable matching method for causal inference in large observational studies. Instead of doing expensive pairwise matching, CRM maps each unit into a 2D representation based on the Mahalanobis distance and the first LDA component. It then matches treated and control units within grid cells in this 2D space. The paper’s main claimed contributions include computational scalability, a pre-matching shortage fraction diagnostic that reports how many treated units lack control support, and a theoretical decomposition of estimation error. Empirically, the authors test CRM on simulations, Criteo, LaLonde, and NHANES, showing that CRM can retain more treated units and scale better than standard matching methods, especially in large datasets. However, the key limitation is that CRM relies on the two-dimensional representation being sufficient for confounding adjustment. This is strong: if confounding is nonlinear, multimodal, or not aligned with the centroid shift, or the distribution of $X|T$ is non-Gaussian, CRM may be biased.

**Additional Comments:**

I believe the paper requires a substantial revision to address the weaknesses.

**Audience:**

No

**Audience Explanation:**

I doubt the scope of the proposed method. The assumptions for LDA and the theorems (in particular, assumption 3) are too strong, and the motivation for using LDA together with the MD is not supported. Moreover, several dimension-reduction-based matching methods have been proposed in the literature, and the paper differs from them only in its use of LDA, which is of limited interest to the audience.

**Claims And Evidence:**

No

**Claims Explanation:**

**Strengths:**
1. The paper introduced LDA to the problem of matching, which has some causal interpretation.
2. Some experiments show the effectiveness of the method when the (strong) assumptions hold.

**Weaknesses:**
1. The presentation is overly dense and hard to follow. The core idea of CRM is relatively simple, but this idea is obscured by a large number of named components, claims, caveats, and terminology.
     * In particular, several phrases such as “bias–balance dissociation,” “self-improving MaxSMD,” and “pre-matching shortage diagnostic” make the manuscript feel more complex than necessary. Moreover, many dash-connected sentences ("--") make the text sound even more AI-generated.
     * Many notations are not defined or defined in different ways, e.g. $k,\sigma(\cdot),\tau_S,\tau(z)$, etc . I urge the author to handle the notation seriously.
2. The motivation of the methodology is unjustified, and the assumptions are too strong. Key related works are missing.
    * Though LDA gives a natural interpretation of the confounding direction, its assumptions are too strong. To make sense, the distribution of $X|T=0,1$ must be Gaussian, and $X|T=1$ must have the same covariance as $X|T=0$.
    * Though the authors argue in Section 4.5 that the LDA weight vector matches the coefficient of a linear treatment assignment model ($\mathbb P(T=1|X) = \sigma(\beta^\top X)$), it does not necessarily mean that for any linear treatment assignment, the underlying distribution of $X|T$ is Gaussian. Therefore, the method can only handle limited cases.
    * Assumption 3 requires that $Z(X) = (d(X),\phi(X))$ satisfies sufficiency, which acts as the foundation for some theorems. Unfortunately, this new assumption is not verified, and the author acknowledged that it is strong, since it compresses all the information of the covariates into a 2D plane. Violation of this assumption could invalidate subsequent theorems.
    * Using the first LDA component as a metric to construct the neighborhood could be understood, but why do you choose the Mahalanobis distance (MD) for the other metric? Note that the MD is regarding the center, so if two units have the same MD, the actual (Euclidean) distance between them could be very large. This means that the units could not be understood as neighbors. Why doesn't the author consider using the first and the second LDA components to construct the cubes? In general, more LDA components could be considered to construct $\phi(X)$.
    * As far as I understood, the introduction of LDA helps to reduce dimension and give causal interpretation. But the latter is not complete (as only the first component is used), or necessary (e.g. KNN methods simply match units with their neighbors). Therefore, I believe methods such as PCA could also be used in the same sense, and require much weaker assumptions than LDA. Many of the following theorems will also hold, as it is not specific to a certain distance metric, i.e. as long as the size of the cube induced by the metric vanishes. Therefore, I believe that the choice of the current $\phi(X)$ lacks sufficient motivation and justification.
    * Based on the above comments, I searched for "Dimension reduction + matching" and realized that there are already several related works [1,2] following this idea that have not been discussed or compared in this paper. For example, [2] shares a very similar idea to this paper, using the PCA instead of LDA. Therefore, the novelty of this paper is reduced, and I urge the authors to include these papers.

3. The theoretical contributions are limited, and some claims are hard to follow.
      * Propositions 1 and 2 follow the standard results, but the proofs are too sketchy and should be formalized into mathematical equations. In particular, the rate should be explicitly calculated, and the results in the reference should be clearly stated. Mere "analogous to" is not acceptable.
      * Theorem 1 is trivial but mixes a population estimand statement regarding $\mathbb E$ with an empirical support set $S$, making the notation unclear.
      * The author claimed in Section 5.4 that Theorem 2 is one of the main contributions, but I could not understand why a sketched proof is still given here for one of the main theorems. The current proof is hard to follow. For example, the theorem states under Assumptions 1,2 the result holds, but the proof uses Assumption 3. Also, the notation for $\tau_S$ is not a function of $X,Z$, and the notation $\tau(z)$ is not conditioned on $S$, but they are used chaotically in the expression of the support restriction bias. The sampling error is also not clear, e.g. where does the $n_T$ term come from? Why didn't the author elaborate on the derivation?

4. The experimental settings are limited to favor the proposed method.
     * No PCA / SDR matching baseline, despite CRM being a dimension-reduction matching method. CRM is essentially a 2D representation-based matching method. There is some literature on matching using sufficient dimension reduction e.g. [1,2], but none has been discussed and compared.
      * Most simulations are built around centroid shifts, elliptical covariance, and linear outcome models. These settings naturally favor a Mahalanobis + LDA representation. The authors do include one nonlinear confounding failure mode, which is good, but it appears late and mainly confirms that all methods struggle. In addition, the confounding could lie in more than one dimension, but the proposed methodology can only handle 1D confounding, as it only uses the first component of the LDA.
      * A central message of the paper is that MaxSMD may not track causal bias. Yet many headline empirical claims still emphasize lower MaxSMD than PSM, especially in the Criteo benchmark. This is somewhat inconsistent.
      * The composite metric (proposed by the author) CRMSE, combines balance and retention. It is intuitive, but it is also newly introduced and embeds a particular value judgment: one unit of retention loss is treated as comparable to one unit of MaxSMD. However, it is not clear why this must hold. Moreover, this can favor CRM because CRM is explicitly designed to retain more units. PSM or CEM may intentionally sacrifice retention to target a more balanced overlap population. The paper should present more frontier-style comparisons (e.g. by plotting the MaxSMD2-Retention frontiers) rather than a single composite score.

**References:**

[1] Luo, Wei, and Yeying Zhu. "Matching using sufficient dimension reduction for causal inference." Journal of Business & Economic Statistics 38.4 (2020): 888-900.

[2] Brown, Derek W., Timothy A. Myers, and Mitchell J. Machiela. "PCAmatchR: a flexible R package for optimal case–control matching using weighted principal components." Bioinformatics 37.8 (2021): 1178-1181.

**Requested Changes:**

See Weaknesses, regarding writing, motivation, methodology, theory, and experiments.

---

> ### Author Response · Authors · 2026-07-02
> **Response to Reviewer 3Cxg,  Part 1: Novelty and Method**
>
> We thank the reviewer for the careful and challenging critique. We agree that the original presentation did not distinguish CRM clearly enough from dimension-reduction-assisted pairwise matching, and that several assumptions and claims needed sharper scope. We have posted a top-level summary of the major changes; here we respond to the methodological and novelty points.
>
> ## Novelty relative to PCA/SDR matching
>
> **Reviewer comment.** CRM appears similar to PCA/SDR matching, and missing related work weakens novelty.
>
> **Response.** We added Luo and Zhu (2020), Brown et al. (2021), and related PCA/SDR/metric-learning work. We also rewrote the framing to clarify the algorithmic difference. PCA/SDR matchers typically use
> $$X \to R(X) \to \text{pairwise distances} \to \text{NN/optimal matching}.$$
> CRM uses
> $$X \to (d(X),\phi(X)) \to \text{cell assignment} \to \text{stratified control sampling}.$$
> Thus CRM is not PCA preprocessing followed by nearest-neighbor matching. It changes the matching operation itself by replacing global treated-control pairwise search with centroid-referenced stratification and sampling. The pre-matching shortage fraction then reports support in this induced representation before outcomes are used.
>
> This reference-relative design is the novelty we now emphasize. It is especially relevant for large observational datasets, where constructing or searching over all treated-control pairs can be expensive or operationally awkward.
>
> **Where revised.** Abstract; Introduction; Related Work; Method: CRM Representation; PCA comparison table/figure.
>
> ## Mahalanobis radius and LDA interpretation
>
> **Reviewer comment.** Equal Mahalanobis radius does not make two units neighbors; LDA assumptions are strong; a linear treatment model does not imply Gaussian covariates.
>
> **Response.** We now state that $(d,\phi)$ are stratification coordinates, not reconstructed pairwise distances. The radius records position relative to the treated centroid, while $\phi$ records signed position along the whitened treated-control mean-shift direction. Two units with the same radius can still differ along angular directions; this is a limitation of the representation, not a property we rely on as full nearest-neighbor distance.
>
> We also clarified the LDA point. CRM does not require Gaussian covariates to compute $\phi$. Gaussianity and equal covariance only justify the exact LDA or linear-log-odds interpretation. Outside that model, $\phi$ is a supervised mean-separation coordinate, not a universally sufficient propensity score.
>
> **Where revised.** Method: CRM Representation; Supporting Geometric Properties; Appendix: Geometric Results.
>
> ## Representation sufficiency
>
> **Reviewer comment.** Assumption 3 is strong, unverified, and foundational; violation can invalidate the theorems.
>
> **Response.** We agree this is central. The revised paper states upfront that ignorability given $X$ does **not** imply ignorability given $Z=(d,\phi)$. For ATT estimation, the required condition is a mean-exchangeability condition in the representation, not automatic preservation of all information in $X$. We added representation-error terms and failure cases for nonlinear, covariance-driven, and multi-axis confounding. The shortage diagnostic is now described only as a support diagnostic in the chosen representation, not as a test of representation sufficiency.
>
> **Where revised.** Abstract; Introduction; Identification Under Representation; Bias Decomposition; Appendix: When CRM May Underperform.
>
> ## Alternative coordinates and PCA/SDR baselines
>
> **Reviewer comment.** Why not use more LDA components, PCA, or other coordinates? Simulations favored Mahalanobis + LDA settings.
>
> **Response.** We clarified that standard two-group LDA gives one discriminant direction, so a “second LDA component” is not directly available in the binary setting. But the broader point is correct: one direction is not universal. We added PCA-95%+NN baselines and stress tests where CRM can fail. We also discuss extensions using additional directions, PCA/SDR coordinates, covariance-gap directions, and learned anchors. The revised paper no longer claims that the default $(d,\phi)$ representation is generally sufficient.
>
> **Where revised.** Ablation Study; PCA-Based Matching; Future Extensions; PCA Sensitivity; Failure Modes.

---

> > ### Author Response · Authors · 2026-07-02
> > **Response to Reviewer 3Cxg, Part 2: Theory, Experiments, and Writing**
> >
> > This comment continues our response to Reviewer 3Cxg, focusing on theory, experiments, metrics, and writing.
> >
> > ## Theory
> >
> > **Reviewer comment.** The propositions/theorems were too sketchy; assumptions, support-set notation, and stochastic terms were unclear.
> >
> > **Response.** We rewrote the theory to avoid informal analogy and overstatement. The revised paper separates four sources of error: representation error, support restriction, discretization error, and stochastic error. We removed “analogous to” proof language and give explicit cell-diameter and occupancy conditions for the fixed-dimension histogram rate.
> >
> > We also separated population support from empirical retained sets. The decomposition now states the representation-sufficiency condition where it is used, writes residual confounding as a function of $Z$, and presents the stochastic term as a conservative order bound rather than an unexplained expression. We no longer describe an $O(n_T^{-1/2})$ MSE bound as a root-$n$ estimator rate.
> >
> > **Where revised.** Estimand Gap; Bias Decomposition; Conservative Rate Bound; Appendix: Proofs.
> >
> > ## Experiments and Criteo
> >
> > **Reviewer comment.** The experiments favored CRM, lacked PCA/SDR baselines, and Criteo emphasized MaxSMD even though balance may not imply low bias.
> >
> > **Response.** We added PCA-95%+NN baselines and stress tests where CRM can fail. We corrected the PSM baseline and reran the simulation/Criteo comparisons. The Criteo full grid now uses collision-free sorted-score PSM. CRM has lower MaxSMD in 31/36 cells and is faster in all 36, but we frame this conservatively: CRM is competitive on balance in this large-scale setting, while its most robust advantage is scalability, high retention, and explicit support diagnostics. We no longer present lower MaxSMD alone as evidence of lower causal bias.
> >
> > **Where revised.** Simulation Results; PCA-Based Matching; Criteo; Discussion.
> >
> > ## CRMSE and frontier reporting
> >
> > **Reviewer comment.** CRMSE embeds a value judgment and can favor CRM because CRM is designed to retain more units. Frontier-style reporting is preferable.
> >
> > **Response.** We agree. CRMSE is now treated only as a descriptive balance-retention summary, not a causal loss function or headline ranking criterion. We report MaxSMD and retention separately and added a balance-retention frontier appendix. The frontier is explicitly framed as a transparency check, not evidence that CRM dominates competing methods.
> >
> > **Where revised.** Simulation Design and Results; main result tables; Appendix: Balance-Retention Frontier.
> >
> > ## Writing, notation, and AI-like prose
> >
> > **Reviewer comment.** The manuscript was overly dense, with too many named components, unclear notation, coined phrases, and dash-connected sentences that sounded AI-generated.
> >
> > **Response.** We agree that the original presentation was too dense. We reorganized the paper into a standard sequence: Introduction, Related Work, Setup, Method, Theory, Experiments, Reporting Standard, Discussion, and Conclusion. We removed coined phrases such as “bias-balance dissociation” and “self-improving MaxSMD,” reduced dash-connected prose, shortened long sentences, and removed promotional language.
> >
> > We also performed a notation audit. Acronyms and symbols are now defined at first use, and the roles of $\pi$, $\hat\pi$, $S$, $S_M$, $\tau_{ATT}$, $\tau_S$, and $\tau_{S^c}$ are stated explicitly. Proven results, empirical observations, and future extensions are now separated.
> >
> > We used language-model tools only for language refinement, code assistance, and debugging. The earlier draft did not receive enough author-side scrutiny for style and notation. In this revision we manually audited writing, notation, references, equations, figures, and tables to remove awkward phrasing and ensure that the final text reflects the authors’ intended scientific claims.
> >
> > **Where revised.** Global revision across Introduction through Conclusion; proofs and technical details moved to appendices.

---

### Review · Reviewer_FfZ7 · 2026-06-20

**Summary Of Contributions:**

The authors propose the centroid-referenced Mahalanobis matching (CRM) approach for matching treated and control units in causal inference from observational studies. The CRM approach addresses the high computational demand of matching-based methods by reducing to linear time complexity in the number of samples. The authors provide both theoretical and empirical results demonstrating the utility of CRM in specific use cases.

## Strengths
- Computation time that is linear in the number of samples $n$, unlike nearest-neighbor matching-based methods that have $O(n^2)$ time complexity.
- Very clear statements of the claims and evidence: which claims have theorems and proofs, which claims have empirical support, and which claims are conjectures supported by intuition.
- Lots of practical considerations are provided, which should be helpful to practitioners making use of the proposed CRM approach.

## Weaknesses
- Organization of the paper is somewhat unusual, with discussion of the proposed CRM approach mixed in with related work and some empirical results mixed in with theoretical results. Despite the unusual organization, the overall quality of writing is acceptable in my opinion.
- Lots of terms and acronyms are not defined and may not be obvious to a reader who is not already familiar with the causal inference literature.
- Modest performance compared to existing methods on accuracy and balance metrics. The only clear improvement is in computation time.

**Additional Comments:**

I didn't carefully review the theoretical results and cannot attest to the correctness of the theoretical claims.

**Audience:**

Yes

**Audience Explanation:**

Yes, causal inference is a topic of great interest to the ML community.

**Claims And Evidence:**

Yes

**Claims Explanation:**

Mostly, with a few experimental results that could be improved.

**Requested Changes:**

Major:
- Bottom of page 5: What is $k$? $k$ seems to be used in later sections as an iterator over the different cells, but it seems to mean something different here. I see that $k=1$ means that $\phi$ is included, but I don't understand what $k=1$ means. $k=2$ is discussed in Section 10.2, and I don't know what that means either.
- Sections 2.4 to 2.6 describe the relation of this paper to the topics discussed in Sections 2.1 to 2.3. I think it would make more sense to make Sections 2.4 to 2.6 subsections or labeled paragraphs within Sections 2.1 to 2.3 so that the relation is clear. Another solution may be to move these discussions to a subsection in Section 4 after discussing the proposed CRM method, similar to the current Section 4.5.
- The MaxSMD metric is never defined, with only a reference provided. Since this is a key metric used for evaluation, and one that may also not be well known, it should be defined in this paper.
- The claimed advantage of CRM over PSM on runtime scaling in Figure 4 is difficult to see. It appears that the quadratic matching should dominate for small p, and I think I can see the runtime curving upwards in the left figure. Increasing $n_T$ by one more data point would help to make this claimed advantage clearer.
- Table 2 inconsistencies: S1 and S3 have multiple methods indicated as lowest CRMSE by the asterisk. The caption also states that $n_t=5,000$ EB results are not shown, but they appear to be in the table.

Minor:
- Lots of acronyms and abbreviations, such as CEM, MaxSMD, and ATT are undefined in the abstract. Consider replacing the sentences with ones that do not introduce new acronyms.
- No citations for CEM and FLAME in second paragraph of intro. No citation for MaxSMD in (C2) in intro.

---

> ### Author Response · Authors · 2026-07-02
> **Response to Reviewer FfZ7**
>
> We have posted a top-level summary of the major changes. Below we respond to the specific points raised in this review. We thank the reviewer for the constructive comments.
>
> ## Notation and acronyms
>
> **Reviewer comment.** The notation $k$ was confusing, and several acronyms/terms were undefined.
>
> **Response.** We revised the notation to avoid overloading $k$. CRM variants are now described in words: radial-only CRM, CRM with the Fisher coordinate, CRM-NN/RunFast. Cell indices are kept separate from variant labels. We also audited first-use definitions of ATT, MSE, MaxSMD, CEM, FLAME, and related acronyms.
>
> **Where revised.** Abstract; Setup and Notation; Method: CRM Representation and CRM Variants.
>
> ## Organization
>
> **Reviewer comment.** Related work, method discussion, and empirical results were mixed in an unusual order.
>
> **Response.** We treated this as a signal that the paper needed a more conventional reading path, not only local edits. We reorganized the manuscript into Introduction, Related Work, Setup, Method, Theory, Simulation, Criteo, NHANES, Reporting Standard, Discussion, and Conclusion. Empirical results were moved out of the method/theory presentation.
>
> **Where revised.** Global restructuring across Related Work, Method, Theory, and experiments.
>
> ## MaxSMD definition
>
> **Reviewer comment.** MaxSMD was not defined.
>
> **Response.** We now define SMD and MaxSMD before using them. For covariate $j$, the standardized mean difference is $SMD_j=|\bar X_{T,j}-\bar X_{C,j}|/\sqrt{(s^2_{T,j}+s^2_{C,j})/2}$, and $MaxSMD=\max_j SMD_j$. We also report MaxSMD and retention separately in all main tables.
>
> **Where revised.** Simulation: Design; main simulation, Criteo, and NHANES tables.
>
> ## Table/caption issues and citations
>
> **Reviewer comment.** Table 2 had inconsistent asterisks/caption wording; CEM, FLAME, and MaxSMD citations were missing.
>
> **Response.** We corrected the table captions and asterisk logic, clarified entropy-balancing notation and retention, and audited all figure/table cross-references. We also expanded Related Work with citations for CEM, FLAME, MaxSMD/balance diagnostics, and balance-frontier perspectives.
>
> **Where revised.** Simulation tables/captions; Related Work; Appendix: Balance-Retention Frontier.
>
> ## Runtime experiment
>
> **Reviewer comment.** The runtime advantage over PSM was hard to see; adding another larger-$n_T$ point would help.
>
> **Response.** We extended the runtime experiment and clarified that runtime comparisons are implementation benchmarks, not lower bounds for all possible implementations. The revised paper separates statistical performance from computational performance.
>
> **Where revised.** Large-Scale Criteo Benchmark; runtime figure; CRM-NN timing appendix.
>
> ## Modest balance/accuracy gains
>
> **Reviewer comment.** CRM’s clearest improvement is computation time; accuracy and balance gains are more modest.
>
> **Response.** We revised the empirical framing accordingly. The paper no longer presents CRM as universally better on finite-sample balance or accuracy. Moderate-size simulations acknowledge that corrected PSM and weighting baselines can have stronger balance-retention tradeoffs. CRM’s contribution is framed as scalability, explicit support diagnostics, high retention in large samples, and competitive balance when pairwise methods become costly.
>
> **Where revised.** Abstract; Simulation Results; Criteo; Discussion; Conclusion.

---

> > ### Comment · Reviewer_FfZ7 · 2026-07-18
> >
> > The authors have made a major revision to the paper that significantly improves upon the presentation. Remaining concerns that can be addressed in a minor revision:
> > - Page 4, "Thus $k = 1$ is a deliberately simple default for settings...": This sentence is placed before the definition of $k$ in the following paragraph. I suggest leaving out $k=1$ and describing the variant in words. Additionally, $k$ still appears to be used as an iterator, but the confusion is less now with the explanation in the "CRM summary" paragraph.
> > - Page 15, "⋆ lowest CRMSE in each scenario-$n_T$ column.": Add "denotes" after the asterisk. I initially thought it was a multiplication since it comes after an equation.

---

### Author Response · Authors · 2026-07-02
**For all reviewers and AE, Summary of major revisions**

We thank the reviewers and action editor for the careful assessment of the paper. The revised manuscript makes substantial changes in organization, framing, theory, experiments, reporting, and uncertainty diagnostics. We have tried to preserve the central contribution while narrowing claims that were too broad in the original version. Although the revision adds several experiments and appendices, we kept the original manuscript's main logic and most of its core content intact so that the revised paper remains continuous with the submitted version. Most new material, including sensitivity analyses, inference diagnostics, and supplementary experiments, is placed in the appendix to keep the main text focused.

**Summary of major revisions.**
The main changes are:

- We reorganized the paper into a conventional sequence: Introduction, Related Work, Setup, Method, Theory, Simulation, Criteo, NHANES, Reporting Standard, Discussion, and Conclusion.

- We rewrote the abstract and introduction to state more precisely what CRM is and what it is not. In particular, CRM is not PCA preprocessing followed by nearest-neighbor matching; it is a centroid-referenced stratification and sampling design.

- We clarified the identification condition. Ignorability given the full covariates $X$ does not imply ignorability after replacing $X$ by $Z(X)=(d(X),\phi(X))$. The revised paper states representation sufficiency explicitly and treats it as an additional assumption.

- We added direct comparisons with PCA-based matching, clarified the relationship to SDR and metric-learning matchers, and reported runtime, balance, retention, and estimation error.

- We corrected the propensity-score matching implementation and reran the simulation and Criteo comparisons using a collision-free sorted-score matcher.

- We replaced the previous NHANES analysis with a real NHANES 2017--2018 extract, added survey-weight sensitivity using `WTMEC2YR`, and moved the synthetic overlap-barrier example to a clearly labeled semi-synthetic stress test.

- We added IPW/AIPW sensitivity results and discussed cross-fitted double/debiased machine learning as an estimation-stage complement.

- We corrected and expanded the theoretical statements, including the support-restriction decomposition, representation error, and the fixed-dimension histogram rate bound.

- In response to the follow-up inference concern, we added a new uncertainty diagnostic. The revised paper now states more clearly that CRM bootstrap intervals are approximate uncertainty summaries rather than guaranteed nominal confidence intervals. We added a conservative cell-stratified standard-error calculation, outcome-model bias-correction diagnostics, and a coverage table separating variance calibration from estimator bias.

- We now report $\hat{\pi}$, final retention, MaxSMD, ATT estimates, uncertainty, and runtime separately. CRMSE is treated only as a descriptive summary, not as a causal loss function.

- We audited notation, acronyms, references, figure/table cross-references, and writing style.

To make the response easy to check, we refer to locations by section, table, and figure number rather than by line number. Exact line numbers can shift when the PDF is regenerated, but these manuscript anchors are stable. Within each reviewer section, we summarize the relevant comment before giving our response.